# FORECASTING WITH LLMS: A DATASET FOR RAPID BACKTESTING WITHOUT TEMPORAL CONTAMINATION

## ABSTRACT

The rise of large language models (LLMs) has made scalable forecasting increasingly feasible, as these models have access to massive amounts of context. Yet evaluating their forecasting ability presents three methodological challenges. Standard benchmarks are vulnerable to *temporal contamination*, where outcomes are already known before the model's training cutoff, and to *staleness confounds*, where newer models gain an advantage from fresher data. Dynamic benchmarks address temporal leakage by tracking unresolved questions, but this results in *long evaluation delays*, since evaluators must wait for outcomes to resolve before judging the accuracy. We address these issues with a forward-only, backtestable forecasting evaluation framework built on frozen context snapshots: contemporaneous, structured summaries of web search results paired with forecasting questions. Our pipeline continuously scrapes unresolved questions from prediction markets and captures their supporting context at the time of scraping, eliminating temporal contamination and mitigating staleness effects. Once questions resolve, these snapshots enable rapid backtesting of diverse forecasting strategies, substantially accelerating research cycles. This framework provides a rigorous, reproducible, and open-source foundation for studying the forecasting capabilities of LLMs. Through two experiments, we demonstrate that our approach enables the rapid identification of effective forecasting strategies.

## 1 INTRODUCTION

Forecasting future events requires reasoning under uncertainty and the timely use of external information (Tetlock & Gardner, 2016). The rise of large language models (LLMs) has made scalable forecasting increasingly feasible, as these models have access to massive amounts of context (Schoenegger et al., 2025; Tan et al., 2024; Schoenegger & Park, 2023; Halawi et al., 2024). Yet evaluating their forecasting ability presents a series of methodological challenges. A central issue is temporal contamination: when models are tested on events occurring before their training cutoff, it becomes unclear whether they are reasoning about the future or simply echoing past information (Lopez-Lira et al., 2025). Another challenge is the staleness confound: models trained on more recent data may appear superior not because of intrinsic forecasting ability, but because their training includes fresher information—even if the event being forecast has not yet occurred.

Traditional benchmarks exacerbate these issues. Static datasets quickly become outdated as new models are trained on more recent corpora. Continuously updated, forward-looking benchmarks that collect unresolved questions reduce leakage but introduce long delays, since evaluation must wait until outcomes are resolved. To address these challenges, we introduce a forward-only, backtestable evaluation framework based on frozen **context snapshots**: contemporaneous, structured summaries of web search results paired with forecasting questions (See Figure 1). Our pipeline continuously scrapes active, unresolved questions and captures the corresponding context at multiple timepoints between the market's initial inclusion in the dataset and its eventual resolution. Because these snapshots are fixed when collected, they eliminate temporal contamination and help control for staleness, increasing the fairness of comparisons across models with different cutoff dates. Importantly, once questions resolve, these frozen snapshots enable replicable, rapid, and efficient evaluation, allowing researchers to test diverse forecasting strategies without waiting for real-world outcomes. Our

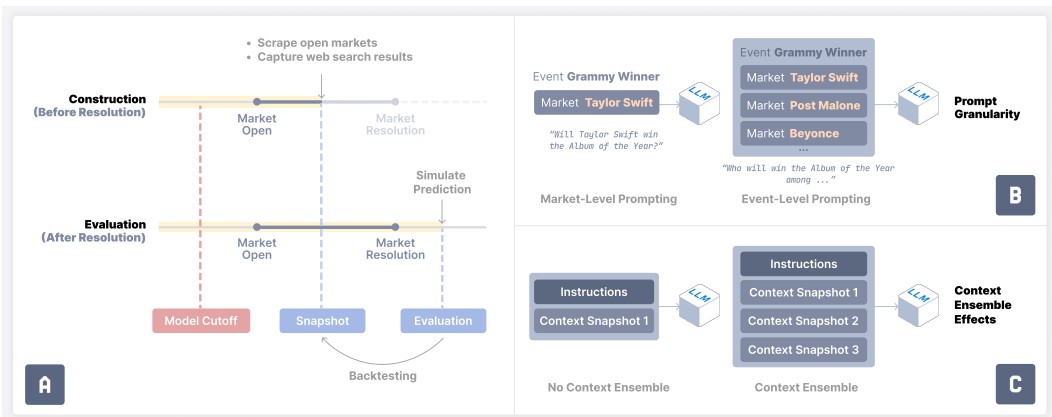

Figure 1: (A) By summarizing the results of web searches conducted before an event resolves, our dataset provides models access to event-relevant information without risking temporal contamination. This dataset can then be used to evaluate models with a knowledge cutoff preceding the event (and systems composed of such models) by providing context snapshots as supporting input. (B) An *event* is a single outcome around which a prediction market is formed, such as "who will win Album of the Year?", or whether the Federal Reserve will raise its benchmark interest rate in a given month. For events with a known set of options, binary ("yes/no") markets are constructed for each option; for events with continuous outcomes, binary markets are typically constructed over intervals of the outcome variable. To generate a forecast, a model can be prompted to make a prediction about a single option without knowledge of the other options (i.e., market-level prompting), or prompted to choose among the full set of options (i.e., event-level prompting.) (C) For each event in the dataset, we generate multiple context snapshots between the market creation and resolution. Models can be prompted with one or more of these context snapshots when making forecasts.

framework thus provides a rigorous, open-source foundation for studying the forecasting capabilities of LLMs, accelerating the development of robust forecasting strategies.

Our initial benchmark consists of 9,388 forecasting questions sourced from a leading prediction market. Of these, 3,338 are questions that, as of the time of writing, have been resolved and include at least one context snapshot, making them immediately available for evaluation. The remaining 6,050 are still active, and we are collecting context snapshots on them and on new questions being launched. The earliest of these context snapshots was taken on July 21, 2025, which falls after the knowledge cutoff of `gpt-5` (September 30, 2024) and other contemporaneous frontier LLMs. As these models' knowledge cut-offs advance, some questions and snapshots will inevitably become outdated. However, our dataset is continuously refreshed by the addition and resolution of new questions, which provide fresh, evaluable snapshots over time. The dataset and code are available at `https://anonymous.4open.science/r/backtest-forecast-2736`.

Our context snapshot scraping pipeline employs two complementary methods for information retrieval. The first method uses a search-integrated language model, specifically `gpt-4o` with Grounding with Bing, which performs live web searches and generates summaries based on the search results. The second method is a custom retrieval-augmented generation (RAG) pipeline. In this approach, we first use `gpt-4o-mini` to generate relevant search queries. These queries are then sent to Dux Distributed Global Search (DDGS) to identify relevant URLs. The content from the retrieved URLs is scraped and subsequently summarized using `gpt-4o-mini`. Finally, we apply a post-hoc filtering step to eliminate summaries that are clearly unrelated to the event in question or that contain leakage.

Through two experiments, we demonstrate the utility of our dataset in enabling the rapid identification of effective forecasting strategies. The first experiment shows that event-level prompting outperforms market-level prompting when using `gpt-4o`, but this advantage does not persist with `gpt-5`. This finding underscores the model-specific nature of forecasting strategies and emphasizes the critical role of backtesting in uncovering the nuanced interactions between a given forecasting strategy and the model used to execute it. The second experiment investigates the effects of ensem-

bling multiple context snapshots, comparing the performance of combining four distinct snapshots against using a single context snapshot. Together, these instances underline the value of backtesting as a tool for distinguishing between key factors that drive predictive performance and those that are less consequential, ultimately guiding the systematic identification of reliable forecasting strategies.

Our framework not only helps mitigate key evaluation pitfalls like temporal contamination and staleness but also enables rapid, reproducible assessment of LLM forecasting. By combining dynamic questions with fixed-time context snapshots, it lays the groundwork for fair and forward-looking evaluation of predictive reasoning in language models.

## 2 RELATED WORK

### 2.1 FORECASTING WITH LLMS

AI systems with forecasting capabilities have significant potential to enhance human decision-making (Hendrycks et al., 2021; Schoenegger et al., 2025). However, prior efforts to evaluate the forecasting abilities of LLMs, such as simulating predictions of historical economic indicators (Hansen et al., 2024), are susceptible to temporal leakage and retrieval contamination, where the information being predicted is already available in the model's pretrained data or supporting context it retrieves through web search (Lopez-Lira et al., 2025; Magar & Schwartz, 2022). In essence, when LLMs are tested on questions whose outcomes were already known prior to the model's knowledge cutoff, it becomes unclear whether the model is genuinely reasoning about the future or merely echoing seen information (Paleka et al., 2025). To overcome these limitations, we introduce a dataset construction pipeline that scrapes active, unresolved questions from live prediction markets and captures contemporaneous web search results at the time of scraping. Because these questions are posted after the model's training cutoff, this approach effectively mitigates contamination risks. Moreover, this design enables rigorous backtesting and efficient evaluation of model forecasts once the associated markets have resolved.

### 2.2 INFORMATION RETRIEVAL FOR FORECASTING

Recent studies have investigated the use of information retrieval to forecast resolved questions. For instance, Pratt et al. (2024) retrieve news articles via the New York Times and Hacker News APIs. However, since these APIs do not preserve historical snapshots, some articles may be retrospectively updated, introducing a risk of temporal contamination. Publishers often update articles using the same URL, making it difficult to ensure that these sources do not contribute to such leakage. Halawi et al. (2024) rely on articles sourced from proprietary third-party news aggregators. However, these closed resources (e.g., GNews, NewsCatcher) do not ensure article permanence, meaning the content may change or become inaccessible over time. In contrast, our dataset is both fully guaranteed to be frozen and entirely open-source, making it readily accessible to the public for research and development purposes. Similarly, Yan et al. (2023) explore forecasting with resolved questions by leveraging Common Crawl. While such web archives provide openly available data, their coverage is often sparse and inconsistent, limiting their utility for fine-grained or time-sensitive forecasting tasks. Appendix A.1 details our empirical experience and observations regarding the limitations of these approaches.

### 2.3 FORECASTING BENCHMARKS

Traditional benchmarks rely on static question sets, which quickly become outdated as modern models are trained on increasingly recent data (Guan et al., 2024; Nako & Jatowt, 2025; Jin et al., 2020; Zou et al., 2022). To address this limitation, recent work has proposed dynamic benchmarks—collections of forecasting questions that evolve over time and are designed to avoid data leakage by including only questions about unresolved future events (Together.ai, 2025; Karger et al., 2024; Zeng et al., 2025). However, dynamic benchmarks face a fundamental challenge: the absence of ground truth at the time of prediction introduces an evaluation delay—the time between making a forecast and the event's resolution, during which accuracy remains unknowable.

Bench to the Future (Wildman et al., 2025) proposes a backtesting framework using archived web crawls to evaluate forecasts on resolved questions. Nevertheless, this benchmark is limited in that it

Table 1: Comparison of recent forecasting benchmarks and our dataset

| Benchmark Name | Dynamic | No Temporal Leakage | No Evaluation Delay | Retrieval Snapshots | Question Count |
|---|---|---|---|---|---|
| ForecastBench (Karger et al., 2024) | ✓ | ✓ | - | - | 6,402 |
| FutureBench (Together.ai, 2025) | ✓ | ✓ | - | - | 42 |
| FutureX (Zeng et al., 2025) | ✓ | ✓ | - | - | ~500 / week |
| Bench to the Future (Wildman et al., 2025) | - | - | ✓ | ✓ | 299 |
| **Our Benchmark** | ✓ | ✓ | ✓ | ✓ | 9,388 (~100 / day) |

relies on Google queries to surface previously crawled pages, introducing noise and possible leakage (See Appendix A.1.4). Moreover, it is closed-source, static, and limited to 299 questions. In contrast, our benchmark is fully open-source, dynamic, and backtestable, fully eliminating retrospective contamination and offering thousands of forecasting questions paired with context snapshots.

# 3 VALUE OF CONTEXT SNAPSHOTS

## 3.1 REDUCING STALENESS CONFOUNDS

Direct comparisons between models with different training cutoffs are inherently confounded by both model quality and staleness. A model trained more recently benefits from fresher information in its training corpus, so an observed performance gap may reflect not only underlying capability but also recency of training data. Without accounting for this confound, such comparisons risk being misleading.

Context snapshots help mitigate this issue by standardizing the information provided to each model, thereby helping to better isolate model performance from the potential influence of training data freshness. Appendix A.2 illustrates this point by comparing `gpt-4.1-mini` and `gpt-4o`, which differ in knowledge cutoff dates.

## 3.2 DECOUPLING MODEL AND SEARCH QUALITY

Comparing two models that both include built-in search capabilities poses a fundamental challenge because model quality and search quality are confounded. Search systems may vary in terms of indexing strategies, coverage, and the freshness of their results. When each model relies on its own retrieval pipeline, it becomes impossible to determine whether any observed performance differences stem from the underlying models or from the quality of the search.

Our frozen context snapshots help address this challenge by holding the retrieval constant. Importantly, this does not imply that information-seeking is unimportant for real-world forecasting; rather, it reflects a deliberate evaluation constraint. By disentangling model performance from search quality, we ensure that differences in forecasting accuracy arise solely from how models reason over the same historical context. Consequently, our dataset supports comparisons of forecasting strategies (e.g., base-rate methods, reference-class approaches) given fixed contemporaneous information, not comparisons of alternative retrieval pipelines or search systems.

It is important to clarify the context our benchmark was designed to address. With regards to the model used for forecasting, this benchmark is designed to evaluate systems based on LLMs that (a) have training data cutoffs that precede the events used for benchmarking; and (b) do not access the live web during inference, which would introduce temporal contamination and invalidate the key advantage of this dataset. While models that are specifically trained to search for forecasting-relevant information may possibly improve forecasting performance, they lie outside the intended scope of systems to be evaluated by this benchmark.

## 3.3 EFFICIENT AND RAPID EVALUATION

Evaluating large numbers of decision strategies is impractical if one must wait for real-world outcomes to unfold. For instance, the question "Will Waymo operate in Las Vegas before Sep 2025?" has a snapshot in our dataset on August 2, 2025, and was resolved on September 1, 2025. Under existing benchmarks, testing this question would require waiting an entire month for the outcome. In contrast, our dataset allows rapid backtesting: one can replicate the LLM forecast relying only on information available on August 2, 2025, and immediately compare it against the eventual resolution. And while a one-month delay may seem manageable, many questions—especially in domains like politics—can take a year or more to resolve, making traditional evaluation painfully slow.

Without such archival snapshots, testing dozens of strategies would take months or even years, as each trial depends on the natural pace of event resolution. Our forward-only benchmark overcomes this barrier by enabling researchers to replay strategies against the same frozen timeline, drastically accelerating evaluation. This design shortens the cycle between experimentation and results, supports repeated evaluations under consistent conditions, and allows for rapid iteration across a wide variety of forecasting strategies. In Section 5, we demonstrate this advantage with two instances of experimentation across various forecasting approaches.

## 3.4 DATA DISTRIBUTION CONSTRAINTS

Direct redistribution of copyrighted articles or web content is typically prohibited due to intellectual property restrictions. In contrast, our context snapshots are structured summaries generated using a search-integrated LLM or a custom retrieval-augmented generation (RAG) architecture powered by a web search library. This approach mitigates legal risks while ensuring reproducibility. From both legal and practical perspectives, structured summarization offers a robust alternative to direct content redistribution.

# 4 BENCHMARK AND DATASET

## 4.1 FORECASTING QUESTIONS

Table 2: The total count of forecasting questions in the current release of our dataset, with their current status, whether active or resolved, recorded as of September 22, 2025.

|  | All Questions | Active Questions | Resolved Questions |
|---|---|---|---|
| **Total** | 9,388 | 6,050 | 3,338 |
| Politics | 4,140 | 4,029 | 111 |
| Sports | 3,325 | 766 | 2,559 |
| Entertainment | 682 | 317 | 365 |
| Science & Technology | 347 | 305 | 42 |
| Finance | 311 | 219 | 92 |
| Economics | 194 | 139 | 55 |
| Climate & Weather | 170 | 113 | 57 |
| Health | 51 | 49 | 2 |
| Other | 168 | 113 | 55 |

Our initial benchmark comprises 9,388 forecasting questions sourced from Kalshi, a leading prediction market. Of these, 3,338 are "backtestable" questions, meaning they have been resolved and include at least one associated context snapshot, making them immediately suitable for evaluation. Each question corresponds to a market on Kalshi.

The earliest snapshot was captured on July 21, 2025, a date that falls beyond the knowledge cutoffs of most frontier LLMs, which generally range from mid-2024 to early 2025. As the knowledge cutoffs of these models continue to advance, some of these snapshots will eventually become outdated. However, this limitation is counterbalanced by the ongoing resolution of new markets, which continuously introduces fresh, evaluable snapshots into the dataset. Our pipeline is designed to actively scrape unresolved questions and capture their supporting context at the time, ensuring that tempo-

ral contamination is avoided. Once these questions are resolved, the frozen context snapshots are dynamically added to the backtestable dataset.

All questions in our dataset are binary, with responses limited to "Yes" or "No." Table 2 provides a detailed breakdown of the total number of questions, along with the distribution of questions across domains. While politics dominates as the most prevalent domain for all questions, the Sports domain leads in terms of resolved questions. This trend can be attributed to the typically short-term, event-driven nature of sports-related questions, which often resolve more quickly compared to other domains. However, as the dataset matures, more and more events from other categories will resolve. Examples of forecasting questions in our dataset are provided in Appendix A.3.

### 4.2 Context Snapshots

Our context snapshot scraping pipeline leverages two independent methods for information retrieval.

The first approach uses a search-integrated LLM, specifically `gpt-4o` with Grounding with Bing Microsoft. This approach conducts live web searches and generates contextual summaries based on real-time search results. This enables dynamic, up-to-date information synthesis directly from the web. On average, we obtained 4.08 summaries per event per date for resolved questions. In total, 4,912 backtestable context snapshots were collected, allowing for backtesting across 1,435 resolved questions. The detailed numerical breakdown is provided in Appendix Table 4. These context snapshots will be referred to as **Bing snapshots** in the subsequent sections.

The second approach leverages a custom retrieval-augmented generation (RAG) pipeline. Initially, `gpt-4o-mini` is employed to generate six search queries based on the details of the provided event. These queries are subsequently fed into the Dux Distributed Global Search (DDGS) library, which returns a set of relevant URLs. The pipeline then scrapes the content from these URLs and utilizes `gpt-4o-mini` to generate concise summaries of the extracted information. Each query yields one summary, resulting in a total of six distinct summaries per event. To ensure relevance, we apply a post-hoc filtering step to eliminate summaries that are clearly unrelated to the event in question. On average, this results in 5.11 relevant summaries per event per date for resolved questions. As a result, our dataset consists of 26,388 backtestable context snapshots generated by the custom RAG pipeline, facilitating forecasts across 2,072 resolved questions. These context snapshots are referred to as **RAG snapshots** in the following sections.

Taken together, these methods provide complementary strengths. Bing snapshots are produced through a high-performing but largely black-box commercial system, whereas RAG snapshots offer greater transparency and customizability at lower cost, albeit with potentially more variability in quality.

To ensure relevant context is captured, we continuously collect snapshots of unresolved forecasting questions through these two methods. For Bing snapshots, 100 questions are randomly sampled from the question pool each day, with six summaries generated for each question. For RAG snapshots, six summaries are generated for every question in the pool. This pipeline is executed on a daily basis through a combination of GitHub Actions and local cron jobs, ensuring the snapshots and questions are consistently timestamped at the moment of scraping. Examples of context snapshots are provided in Appendix A.5.

### 4.3 Market Price Snapshots

Our pipeline also captures the daily market prices associated with the forecasting questions in our database. These prices represent the aggregated beliefs of human participants, serving as a valuable baseline for comparison. To calculate the market price, we divide the "yes" price by the sum of the "yes" and "no" prices, providing a normalized measure of collective human judgment.

## 5 Experiments

In this section, we demonstrate two instantiations of backtesting applied to different forecasting strategies. The primary objective is to show that our dataset supports rapid iteration and evaluation of a set of strategies. We do not intend to claim that any single strategy significantly improves

forecasting accuracy. Rather, these experiments serve as demonstrations of how our backtestable framework enables rapid evaluation of diverse strategies. The primary contribution of this work is the methodology and dataset that make such backtesting possible, not the magnitude of gains from any particular strategy we happened to test.

## 5.1 METRIC

Since our forecasting questions are binary, we evaluate performance using the Brier score, a standard metric for probabilistic predictions. The Brier score is defined as $(f - o)^2$, where $f \in [0, 1]$ represents the forecasted probability, and $o \in \{0, 1\}$ is the actual outcome. Lower Brier scores indicate better forecasting performance, with a score of 0 representing perfect accuracy. A forecast of 0.5, reflecting complete uncertainty, yields a Brier score of 0.25, serving as a baseline for uninformed predictions.

## 5.2 PROMPT GRANULARITY

### 5.2.1 BACKGROUND

Our dataset is structured hierarchically, consisting of events and their associated markets (or questions). An event (e.g., a political election) can contain one or more markets (e.g., individual candidates). The structure of these markets can vary depending on the nature of the event.

Some events feature mutually exclusive outcomes, where only one market can resolve positively. For example, in a prediction about the winner of an award, only one candidate can win, so only one market can resolve in the affirmative. Other events follow a ladder-style structure, where markets represent incremental thresholds (e.g., predicting whether the temperature will exceed 50°F, 60°F, 70°F, and so on.) In such cases, multiple markets may resolve positively depending on the final outcome, as each threshold is met.

In addition, there are non-mutually exclusive events, where several markets can resolve positively at the same time. An example of this would be predicting which companies will run Super Bowl ads, where multiple companies may be involved, and more than one market could resolve "yes." All context snapshots are generated at the event level, ensuring that they capture the full set of associated markets and the broader framing of the event.

In this experiment, we aim to explore the effectiveness of different prompting strategies by leveraging the event-market hierarchy. Specifically, we compare the impact of event-level prompting, where prompts include event-level data of all options (e.g., including all nominees for a given Oscar award), versus market-level prompting, where forecasting prompts only include information about a single option/outcome (e.g., asking whether a specific actor will win a given Oscar award, without including information on who else is nominated for the same award).

### 5.2.2 EXPERIMENTAL SETUP

**Conditions.** Market-level prompting includes only the metadata for a single market, such as the specific market question (e.g., "Will Taylor Swift win the Grammy Awards?"). In contrast, event-level prompting incorporates the broader context of the entire event, including the event title (e.g., "Who will win the Grammy Awards?"), and the metadata for all associated markets (e.g., a list of nominated candidates). To assess the efficacy of these two strategies, we compare the performance of event-level and market-level prompting, both with and without the inclusion of context snapshots. As a baseline, we also evaluate market prices to gauge model accuracy relative to collective human predictions, offering a point of comparison for the model's performance.

**Models.** We compare these conditions using six models: `gpt-5`, `gpt-4o`, `claude-3.5-haiku`, `gemini-2.0-flash`, `llama-3.1-70B`, and `qwen-2.5-72B`. This design allows us to investigate whether there are model-specific differences in the effectiveness of the prompting strategies and to explore how interactions between each strategy and each model may influence performance.

**Sample.** Our experimental evaluation draws on 1,336 questions, which correspond to 566 unique events. The sampling procedure is detailed in Appendix A.10. For each event, we

use four RAG snapshots. All included markets were published after the respective model knowledge cutoffs—September 30, 2024, for `gpt-5`; October 1, 2023, for `gpt-4o`; July 2024 for `claude-3.5-haiku`; August 2024 for `gemini-2.0-flash`; December 2023 for `llama-3.1-70B`; and late 2023 for `qwen-2.5-72B`. While all markets had been resolved by the time of evaluation, their resolutions occurred after the context snapshots were generated (i.e., after the simulated prediction time).

### 5.2.3 RESULTS

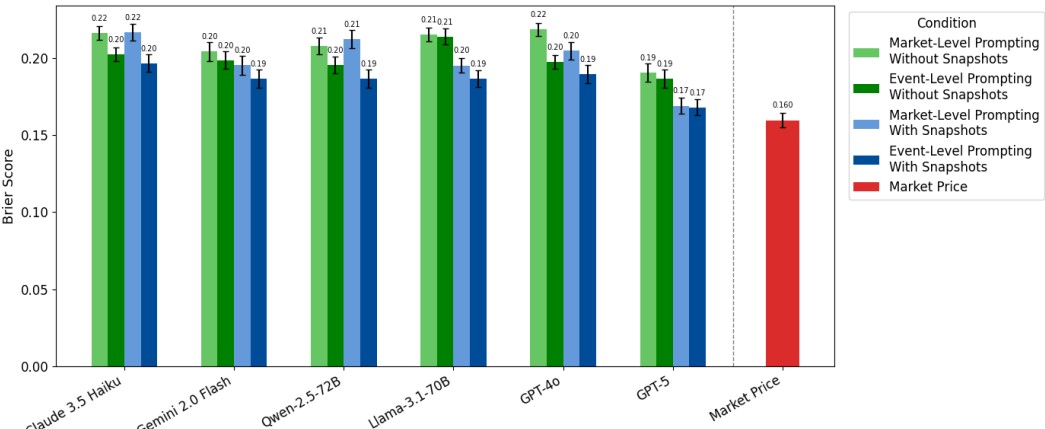

Figure 2: Mean Brier scores across all models under different prompting granularities and context snapshot conditions. Bars compare market-level and event-level prompting, each evaluated with and without context snapshots. Error bars indicate $\pm 1$ standard error of the mean, computed across questions.

Results reveal clear differences in how event-level versus market-level prompting influences model performance. Mixed-effects regression shows that `gpt-4o`, `claude-3.5-haiku`, and `qwen-2.5-72B` experience a significant accuracy drop when using market-level prompting rather than event-level prompting ($p < 0.01$ for all interaction terms). In contrast, for `gpt-5`, `gemini-2.0-flash`, and `llama-3.1-70B`, the distinction between event-level and market-level prompting effectively disappears.

Across all models, the inclusion of context snapshots consistently improves accuracy. The regression shows that removing snapshots increases Brier scores by approximately 0.012 on average ($p < 0.001$), confirming that snapshots provide useful additional information regardless of model or prompting strategy.

Taken together, these findings highlight that the effect of prompt granularity is model-dependent. `gpt-4o`, `claude-3.5-haiku`, and `qwen-2.5-72B` gain a measurable advantage from event-level prompting, whereas `gpt-5`, `gemini-2.0-flash`, and `llama-3.1-70B` show no sensitivity to prompt granularity. Rapid backtesting thus proves valuable in uncovering these model-specific dynamics and clarifying which design choices matter most for different systems. Detailed results of the regression are presented in Appendix A.8.

### 5.3 CONTEXT ENSEMBLE EFFECT

### 5.3.1 BACKGROUND

While prior work has discussed the idea of model ensembling in forecasting tasks (Schoenegger et al., 2024; Karger et al., 2024), much less attention has been paid to ensembling summarized context in the realm of information retrieval. We address this gap by providing the first evidence that ensembling summarized context, by combining multiple context snapshots, can measurably affect the forecasting performance of language models.

We conduct two experiments. In the first experiment, we ensemble Bing snapshots generated via identical processes, each prompted in the same way. In the second experiment, we ensemble snapshots derived from divergent processes—summaries produced by our RAG pipeline, where each is based on a distinct search query.

In both settings, we compare LLM forecasting performance when the model is conditioned on a single snapshot versus multiple snapshots provided together in the prompt.

### 5.3.2 EXPERIMENTAL SETUP

**Conditions.** Each experiment includes three conditions: (1) "Ensemble", where the LLM receives an ensemble of multiple context snapshots; (2) "No ensemble", where only one context snapshot is provided; and (3) "Market price", where the forecasting LLM is provided the market price for the event, which serves as a baseline measure of the benefit gained by knowing aggregated human beliefs.

**Models.** We employ OpenAI's `gpt-5` model for both experiments. Since our objective is to investigate the effects of ensembling context snapshots rather than to compare different models, we use the same model across all experimental settings. We select `gpt-5` because it represents one of the most state-of-the-art language models currently available.

**Sample.** In the first experiment, we use a sample of 779 questions associated with 340 distinct events, each accompanied by four Bing snapshots. For the second experiment, we employ the same sample used in the prompt granularity experiment, consisting of 1,336 questions associated with 566 unique events, each supplemented with four RAG snapshots.

### 5.3.3 RESULTS

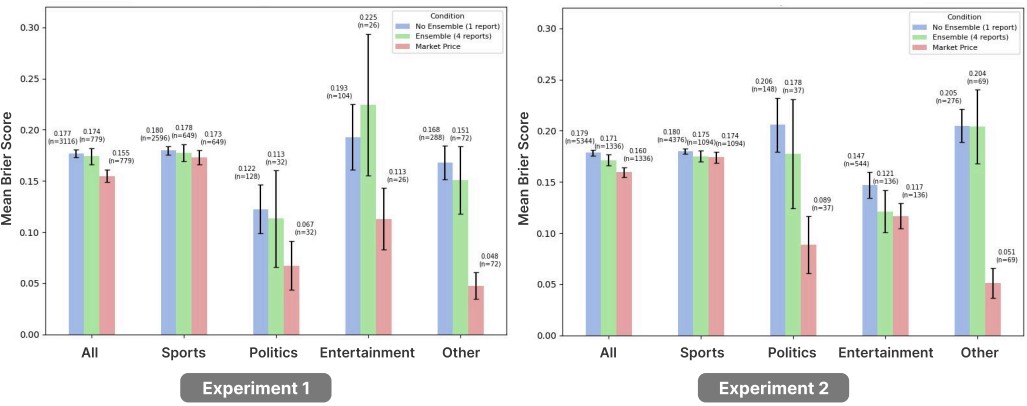

Figure 3: Mean Brier scores by condition and domain

**Experiment 1.** Using multiple context snapshots produced a slight overall improvement relative to a single snapshot, with the mean Brier score dropping from 0.177 to 0.174. By domain, in politics, ensembling reduced the score from 0.122 to 0.113, although market prices remained far stronger at 0.067. Sports saw only a marginal gain (0.180 to 0.178), and in other domains the improvement was somewhat larger (0.168 to 0.151). In contrast, entertainment showed worse performance under ensembling (0.193 to 0.225), though this result is difficult to interpret given the small sample size.

**Experiment 2.** The overall pattern was similar: ensembling slightly improved forecasts across all domains, with the mean Brier score dropping from 0.179 to 0.171. Politics and entertainment benefited most. In politics, ensembling enhanced scores from 0.206 to 0.178, though markets were still far ahead at 0.089. In entertainment, ensembling reduced the score from 0.147 to 0.121—slightly underperforming markets at 0.117. Sports showed a minor improvement (0.180 to 0.175), while in the other domains, ensemble and no-ensemble were nearly identical (0.205 vs. 0.204), with markets much stronger at 0.051.

Together, the two experiments demonstrate that context ensembling may improve LLM forecasts, though the degree of benefit varies considerably by domain. Gains are more pronounced in politics and entertainment, and more modest in sports, and negligible elsewhere. Despite these improvements, market prices remain the strongest baseline overall.

# 6 CONCLUSION

## 6.1 LIMITATIONS

Our work has several limitations. First, while context snapshots help reduce staleness-related confounds, they do not eliminate them entirely. For instance, the models may rely more on their training data than on the context snapshots, compromising the effectiveness of our solution. The ideal solution would be to control the pretrained dataset across models, which is infeasible in practice. As a result, context snapshots serve as a viable alternative. They standardize the information provided to each model to a reasonable degree, helping to alleviate the confounding issues. While not perfect, our approach represents a pragmatic compromise for feasibility.

Second, our structured summaries are not raw data but derived representations. As such, they may omit certain details or nuances that were present at the time of scraping. This is partly due to the inherent limitations of summarization, as well as legal constraints that prevent the open release of raw web content. Furthermore, capturing the full contextual landscape of any given market comprehensively is inherently difficult—if not impossible.

Finally, the dataset is drawn exclusively from a single prediction market, Kalshi. To improve generalizability, we may expand it in future work to include additional platforms. Nevertheless, Kalshi is one of the largest real-money forecasting platforms, characterized by high question volume and broad topical coverage, making it a strong initial testbed. Moreover, most prediction platforms share a similar trading structure to Kalshi, and Kalshi spans a wide range of domains, including sports, politics, economics, and global events, which substantially overlap with other leading prediction markets.

## 6.2 SUMMARY

We introduce a forward-only, backtestable evaluation framework that pairs forecasting questions with frozen context snapshots, enabling fairer and more efficient assessment of LLM forecasting capabilities. Our experiments further demonstrate the value of systematic backtesting: uncovering model-specific differences in prompting strategies and highlighting the benefits of ensembling diverse context snapshots. Together, these contributions advance the study of forecasting as a testbed for reasoning under uncertainty, offering practical utility for the broader research community.

# 7 REPRODUCIBILITY STATEMENT

We have made extensive efforts to ensure the reproducibility of all results presented in this paper. All code, datasets, and the specific event identifiers used in our experiments will be open-sourced under a clear license. Additionally, all prompts used in our experiments are included in Appendix A.9, A.7, and A.6, enabling full independent verification and promoting transparency.

# 8 ETHICS STATEMENT

This work does not involve human subjects or sensitive personal data. All data are derived from publicly available prediction markets and web content, which we summarize into structured context snapshots to mitigate copyright and privacy concerns. Our dataset and code will be released under an open-source license to ensure transparency and reproducibility. While our framework is designed for research on forecasting and reasoning, we recognize that forecasting technologies could be misused for disinformation or manipulation; we therefore encourage responsible use and emphasize that our dataset is intended for scientific study.

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

## A APPENDIX

### A.1 LIMITATIONS OF PRIOR RETRIEVAL METHODS

#### A.1.1 NEW YORK TIMES API

We observed discrepancies among the date sources associated with news articles, including those recorded in the API, those embedded in the web page's metadata (e.g., HTML meta tags), the dates mentioned within the article content itself, and the actual last updated timestamp of the article. For example, consider the article titled *"A Trade Weapon"* available at `https://www.nytimes.com/2025/01/28/briefing/donald-trump-tariffs.html`. As queried on September 22, 2025, the New York Times API returns the `pub_date` field as January 28, 2025, and the corresponding HTML meta tag on the web page also indicates January 28, 2025. However, the article was subsequently updated on March 26, 2025, and includes content and references added after the original publication date. This discrepancy highlights the risk of temporal contamination when relying solely on API-provided publication dates for information retrieval.

### A.1.2 THIRD-PARTY NEWS AGGREGATORS

Proprietary third-party news aggregators, such as GNews and NewsCatcher, do not guarantee content permanence or stability. Moreover, both platforms require costly subscriptions to access their data APIs, further complicating their use for long-term or reliable data tracking.

For instance, consider an article about National Footbal League coach Aaron Glenn, initially published on November 29, 2024, which can be accessed at `https://abc7.com/post/nfl-coaching-changes-2024-latest-firings-openings-rumors/15603353/`. On July 11, 2025, this article was still available through the GNews API. However, by September 23, 2025, querying GNews with the exact title, or even with highly similar terms, yielded no results, even when specific date ranges were applied. This discrepancy suggests that GNews does not guarantee the permanence of articles, and content may be removed or altered over time, leading to potential inconsistencies in data. The underlying cause could be related to limited licensing agreements, which may restrict access to certain content after a period.

Additionally, this article was updated on February 11, 2025, several months after its initial publication, raising the possibility of retrospective changes to the content. If the article's URL is used directly without an API request, there is a risk of incorporating outdated or altered versions of the content, introducing further potential contamination in the data.

As for NewsCatcher, the prohibitively high cost of its subscription has prevented us from verifying whether the integrity of its content is maintained over time. This leaves uncertainty regarding the reliability and consistency of articles sourced through its API.

### A.1.3 COMMON CRAWL AND WAYBACK MACHINE

Although web archives like Common Crawl offer publicly available data, their coverage is often sparse. For example, when querying the URL pattern `"cnn.com/2024/01/*"`, Common Crawl returns only 146 crawls, while the Wayback Machine provides access to 7,050 unique URLs for the same period. Similarly, for the pattern `"nytimes.com/2024/01/*"`, Common Crawl returns no results, while Wayback Machine archives 16,497 unique URLs. The discrepancy is even more stark for the pattern `"variety.com/*"`, where Common Crawl only captures 52 crawls, compared to over 50,000 URLs available on the Wayback Machine. These figures are based on an analysis of 15 Common Crawl indexes spanning 2024 to 2025, underscoring limitations in its temporal coverage.

While the Wayback Machine offers broader archival depth, it presents its own constraints. It is not designed for bulk access, and its API is rate-limited and occasionally inconsistent. Bulk downloading or automated scraping typically requires special permissions. Moreover, the archive primarily provides raw HTML snapshots, which can sometimes be incomplete or corrupted. Critically, neither CommonCrawl nor the Wayback Machine supports keyword-based contextual search, making it impossible to query efficiently. Because these archives require URL-based access, these archives do not readily support a reproducible pipeline for retrieving contemporaneous, thematically relevant material for each forecasting question.

Our framework is best viewed as complementary rather than competitive with large web archives. Researchers who wish to augment our snapshots with additional archival context can do so, since our dataset already provides the questions, queries, and timestamps needed to anchor such searches. Our contribution is not intended to replace these archives, but to offer a clean, leakage-free, forecasting-specific layer on top of the broader information ecosystem.

In short, while CommonCrawl and the Wayback Machine serve as excellent general-purpose archives, they do not resolve the practical reproducibility or legal challenges inherent in large-scale, forward-only forecasting evaluation. Our framework fills that gap.

### A.1.4 GOOGLE SEARCH WITH DATE LIMIT

Limiting Google search to a cutoff date does not fully address temporal contamination because search engines continually re-rank and retroactively update pages. Even if results are filtered by date, snippets, metadata, and page contents frequently reflect information added after the nominal

cutoff, and many pages are undated or incorrectly dated. As a result, date-filtered live search is still exposed to leakage, making the evaluation neither reproducible nor temporally secure.

Bench to the Future (Wildman et al., 2025) acknowledges that using live Google searches introduces the potential for information leakage based on the fact that the order of certain search results likely changed over time.

Frozen snapshots solve these problems by capturing information as it appears at the time of query (which occurs before event resolution), independent of later edits, page removals, or algorithmic changes (which may occur after an event is resolved). This creates a stable and contemporaneous record that supports controlled backtesting.

## A.2   STALENESS CONFOUNDS

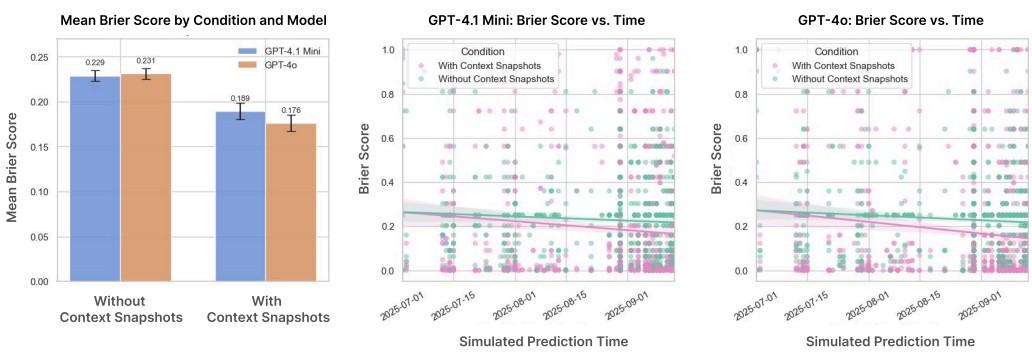

Figure 4: Brier scores over time and by condition for `gpt-4.1-mini` and `gpt-4o`. Left/middle: Each marker represents a forecasting question; solid lines indicate ordinary least squares (OLS) trend lines with 95% confidence intervals (lower is better). When provided with context snapshots, both models exhibit reduced forecasting errors, particularly for questions with later snapshot times, compared to conditions without snapshots.

Comparing models with different training cutoffs is problematic because performance differences may reflect both model quality and the freshness of training data. Newer models benefit from more recent information, skewing results. Context snapshots can help control for this by standardizing inputs, making it easier to assess true model performance.

Figure 4 illustrates this point by comparing `gpt-4.1-mini` and `gpt-4o`, which differ in knowledge cutoff dates. `gpt-4.1-mini` was trained on data up to June 1, 2024, while `gpt-4o`'s cutoff was October 1, 2023. We evaluate both models across simulated prediction dates ranging from July 2025 to September 2025, corresponding to the timestamps when the context snapshots were collected. The evaluation draws on a random sample of 947 questions from our dataset.

In the absence of context snapshots, the mean difference in Brier scores between `gpt-4.1-mini` and `gpt-4o` is negligible. However, once context snapshots are introduced, `gpt-4o` tends to perform slightly better than `gpt-4.1-mini`. This shift could indicate that `gpt-4.1-mini` might have an advantage in settings without snapshots due to its more recent knowledge cutoff. In other words, when both models are provided with the same up-to-date contextual information, the difference in their forecasting abilities becomes noticeable, with `gpt-4o` showing a slight edge.

These findings highlight that LLMs trained on static corpora may grow stale over time. Interestingly, a more recent model, despite having a smaller or less comprehensive training set, can sometimes appear superior, or at least comparable, to an earlier model. This phenomenon often arises because the newer model benefits from more recent data, which may provide an advantage in certain circumstances. However, context snapshots, which provide timely updates to both models, help mitigate this effect. This allows for a relatively fairer comparison that focuses on the intrinsic forecasting capabilities of the models, rather than the mere recency of their training data.

## A.3 EXAMPLE FORECASTING QUESTIONS

Table 3 presents a set of example forecasting questions drawn from various domains within our dataset.

Table 3: Examples of forecasting questions in our dataset

| Category | Market Title | Market Subtitle | Primary Rules |
|---|---|---|---|
| Politics | Will Stacy Garrity be the Republican nominee for Governor in Pennsylvania? | Stacy Garrity | If Stacy Garrity wins the nomination for the Republican Party to contest the 2026 Pennsylvania Governorship, then the market resolves to Yes. |
| Sports | Abilene Christian vs Tulsa Winner? | Tulsa | If Tulsa wins the Abilene Christian vs Tulsa college football game originally scheduled for Aug 30, 2025, then the market resolves to Yes. |
| Entertainment | Will Taylor Swift release a song this month? | Taylor Swift | If Taylor Swift releases a song on Spotify after issuance (August 11, 2025) and before Sep 1, 2025, then the market resolves to Yes. |
| Science & Technology | Best AI at the end of 2025? | ChatGPT | If OpenAI has the top-ranked LLM on Dec 31, 2025, then the market resolves to Yes. |
| Finance | Will Klarna or Stripe IPO first? | Klarna | If Klarna confirms an IPO first, before Jan 1, 2040, then the market resolves to Yes. |
| Economics | When will the next U.S. recession start? | Q4 2024 | If the NBER declares the peak of American business activity predating a recession to be in Q4 2024, then the market resolves to Yes. |
| Climate & Weather | Will it rain in NYC on Sep 12, 2025? | Rain in NYC | If the number of inches of precipitation recorded at Central Park, New York on September 12, 2025 is strictly greater than 0, then the market resolves to Yes. |
| Health | Will English resident doctors strike before Aug 2025? | Before Aug 2025 | If resident doctors in England have engaged in strike action before Aug 1, 2025, then the market resolves to Yes. |
| Other | When will the Amtrak Acela II trains enter revenue service? | Before Aug 1, 2025 | If the Amtrak Acela II trainsets have entered revenue service before Aug 1, 2025, then the market resolves to Yes. |

## A.4 THE TOTAL NUMBER OF CONTEXT SNAPSHOTS

Table 4 presents the total number of context snapshots collected by each method in our dataset, along with their average length and the average number of snapshots per event.

Table 4: The total count of context snapshots per method in our dataset, with their average length and count per event, recorded as of September 22, 2025.

|  | Bing Snapshots | RAG Snapshots |
|---|---|---|
| Backtestable Snapshots | 4,912 | 26,388 |
| Resolved Questions with Snapshots | 1,435 | 2,072 |
| Resolved Events with Snapshots | 609 | 785 |
| Mean Count Per Event Per Date | 4.08 | 5.11 |
| Average Length in Characters | 2427.02 | 2566.88 |

## A.5 EXAMPLE CONTEXT SNAPSHOTS

Table 5 shows an example of a Bing snapshot. Table 6 shows an example of a RAG snapshot. Table 7 shows examples of RAG snapshots from our dataset, paired with the corresponding GPT-5 predictions and reasoning, each produced using only a single snapshot as input.

Table 5: An example of a Bing context snapshot generated with search-integrated LLM.

---

Alexander Shevchenko and Reilly Opelka are set to compete in the first round of Wimbledon Men Singles 2025 (Round of 128) on Tuesday, July 1. Grass court dynamics and player performance history are crucial in evaluating both players' chances.

1. Shevchenko's record on grass is notably weak with limited exposure. He has played just five matches on the surface, with only one victory in 2024. This year, he faced an early exit in Mallorca qualifying after a loss to Elias Ymer. His experience and lack of familiarity with grass court mechanics position him as an underdog. His head-to-head record versus Opelka is also disadvantageous, with a loss in their previous meeting.

2. Reilly Opelka has demonstrated competence on grass despite inconsistent form in 2025. His powerful serve thrives on grass, averaging over 11 aces per match during this year's season. However, he experienced early exits in recent tournaments including Queen's Club and Eastbourne. Nevertheless, he is favored substantially in this matchup considering his higher ATP rank, grass court experience, and his previous victory against Shevchenko.

3. Predictive models and simulations consistently favor Opelka's victory, assigning him a 66-71.4% likelihood of winning against Shevchenko. Shevchenko's odds of advancing are generally in the range of 30-34% based on data analysis by Dimers and Bleacher Nation. Opelka's strong serve, strategic play on grass, and higher match experience are cited as key factors.

4. Betting experts have indicated Opelka as a firm favorite in terms of moneyline odds. His aggressive style and ability to capitalize on pressure points further solidify expectations for his advancement in the tournament. Shevchenko's inexperience and performance inconsistencies, particularly on this surface, amplify his challenges.

5. Summing up, Reilly Opelka's strengths on grass and prior dominance over Shevchenko alongside statistical models mark him as a clear favorite in this Round of 128 match. The likelihood of Alexander Shevchenko pulling off an upset remains low.

---

Table 6: An example of a RAG context snapshot.

---

As Pope Leo XIV approaches the 100th day of his ministry, he has focused on adapting to his new role while preparing for future changes within the Vatican. His first public address emphasized a commitment to a synodal church and the key objectives of his papacy center around the teachings of Vatican II, including growth in collegiality, attention to the faithful, and dialogue with the contemporary world. The article indicates that during this initial phase, the pope holds meetings to understand the structure of Vatican functions before making significant appointments. (August 12, 2025, https://www.usccb.org/news/2025/pope-leos-first-100-days-leaning-his-new-role)

Pope Leo XIV, elected on May 8, 2025, is the first pope from the United States, with a background that includes significant time spent in Peru as both a missionary and bishop. His election is characterized by the melding of American and Latino cultural influences, reflecting his diverse heritage. The article discusses his academic background and pastoral experience, noting his previous role in church governance as the head of the Dicastery for Bishops. It emphasizes his potentially progressive stance on social issues, while also highlighting certain doctrinal conservativisms, such as his opposition to the ordination of women as deacons. (May 10, 2025, https://www.cbsnews.com/news/new-pope-robert-prevost-pope-leo-xiv/)

Pope Leo XIV's election is marked by a focus on continuity with the previous pope, emphasizing the establishment of a dialogue-centered church. This context places him in a difficult position as he takes over during a period of scrutiny regarding clerical sexual abuse and broader calls for social justice. His public statements suggest an intention to honor his predecessor's legacy while potentially carving out a distinct path for his papacy. He is expected to engage with international political issues but must navigate the church's internal challenges, which include ongoing crises stemming from past scandals. (May 8, 2025, https://www.nbcmiami.com/news/national-international/new-pope-who-is-robert-francis-prevost/3610225)

Pope Leo XIV's inaugural address illustrated his aim to foster unity and love within the church while addressing the need to transform it into a more missionary entity. His election, which highlights a departure from the traditional selection of non-Americans for the papacy, reflects a notable shift in the church's dynamics and points towards his potential role in utilizing his American background to engage with contemporary challenges. The article discusses the political implications of his election and how it may resonate with American Catholics. (May 8, 2025, https://www.cnn.com/2025/05/08/europe/new-pope-conclave-white-smoke-vatican-intl)

The election of Pope Leo XIV represents a significant event for the Catholic Church, as it introduces the first American pope. His choice of the name "Leo" is seen as symbolic and connected to his commitment to social reform and legacy continuity with earlier popes. The article mentions his advocacy for marginalized communities and the political challenges he may face regarding issues like migration, human rights, and clerical abuse. Overall, his papacy is positioned at a historically pivotal moment, with expectations for both internal reform and external engagement. (May 8, 2025, https://www.cnn.com/2025/05/08/europe/new-pope-conclave-white-smoke-vatican-intl)

---

Table 7: Examples of context snapshots from our dataset, paired with the corresponding GPT-5 predictions and reasoning. Each prediction was produced using only a single snapshot as input. The first example uses a Bing snapshot, and the second uses a RAG snapshot.

| Question | Context Snapshot | Prediction | Reasoning |
|---|---|---|---|
| How much will core PCE increase in July 2025? | 1. Economists widely agree that the core Personal Consumption Expenditures (PCE) inflation in July 2025 was expected to rise 0.3% month-over-month, slightly higher than the June rate of 0.26%. Tariff-related inflationary pressures and goods price increases contributed to this trend. While core PCE annual rate was predicted to increase to 2.9% from 2.8%, the monthly growth rate appeared steady at 0.3%. Such figures align with the Bureau of Economic Analysis consensus data and economist reviews3:0†source3:1†source3:2†source. 
 2. The underlying inflation model indicated steady month-to-month pressures from tariffs driving the inflation rate. Key factors included higher costs for consumer goods, services, and shelter indexes. Though some economists speculate that the inflation caused by tariffs may eventually stabilize, July's inflation remained above the Federal Reserve target of 2%, reflecting persistent upward trends3:0†source3:2†source. 
 3. Goldman Sachs forecasts for July core inflation also fell within the predicted range of 0.3% monthly increments, projecting a consistent upward spiral driven by external trade policies impacting consumer pricing. Meanwhile, other financial institutions, including BofA, offered similar figures confirming 0.3% growth in core metrics aligned with consensus3:2†source3:3†source. 
 4. Economic reports coming before late August further emphasized robust upward trends in July inflation as driven by national policy factors and post-pandemic residual bracketing between goods/resolution shifts. Bureau Measured PCE review will add attributes finality later impacting PCE Status driving above set evolving resolve next weeks towards median/public dimensions | 0.35 | Above 0.3% requires a 0.4%+ print at one decimal. With consensus at 0.3%, upside tail to 0.4% exists but is not the base case. |
| Will any Texas House Democrat be arrested in Aug 2025? | Texas Democrats defied Republican efforts by not attending a legislative session on August 8, 2025, aimed at voting on a redistricting plan that would increase Republican congressional seats. In response, Texas Attorney General Ken Paxton expressed a willingness to target those who remain absent from the state for contempt and indicated plans for potential civil arrests. While some Democrats expressed a readiness to return pending agreements on various issues, Governor Greg Abbott has pursue lawsuits to remove absent Democrats from their posts and threaten the legitimacy of their seats. This situation resulted in civil warrants being sought for multiple Democrats who left the state to avoid voting on the proposed congressional maps. (ABC News, August 8, 2025, https://abcnews.go.com/Politics/texas-democrats-face-republican-deadline-stop-standoff-face/story?id=124480936) 
 The Texas House took action to issue civil arrest warrants on August 4, 2025, for over 50 Democratic lawmakers who left the state to break quorum in order to block the passage of a new congressional map. The warrants allow for arresting these legislators only within Texas state lines, making them largely symbolic as most absentees relocated to states like Illinois and New York. A similar tactic was previously attempted in 2021, when an injunction was upheld against such warrants, indicating an ongoing legal battle over quorum rules. The House's move signaled an escalating conflict between the parties over redistricting efforts. (Texas Tribune, August 4, 2025, https://www.texastribune.org/2025/08/04/texas-democrats-house-warrants-arrest-quorum-break/) 
 Governor Abbott issued direct orders for the arrest of Texas Democrats on August 4, 2025, after they failed to comply with a deadline to return to the legislative session. He criticized their absence as un-Texan and accused them of abandoning their duties. Abbott outlined potential consequences, suggesting the removal of seats for non-compliance and claiming that their absence was obstructing critical legislation. The legislative standoff has drawn national attention due to its implications on congressional balance and has prompted Abbott to mobilize state authorities to ensure the return of the absent Democrats. (Fox 4 News, August 4, 2025, https://www.fox4news.com/news/abbott-orders-arrest-democrats-legislative-standoff) 
 The ongoing standoff between Texas Republicans and Democrats continued as the Texas House failed to achieve quorum for a third time by August 11, 2025. This notable because a series of Democratic lawmakers had traveled to different states, while Republican officials expressed increasing frustration at their absence from legislative duties. Amid this dynamic, Texas Attorney General Ken Paxton filed lawsuits aiming to remove specific House Democrats from office. The contention largely centers around the potential redistricting plan, which Democrats argue is discriminatory while Republicans assert that Democrats are not fulfilling their legislative responsibilities. (CBS News, August 11, 2025, https://www.cbsnews.com/texas/news/texas-legislative-standoff-continues-lawsuits-state-house-democrats-protesting-redistricting-proposal/) 
 On August 4, 2025, the Texas House voted to track down and arrest Democratic lawmakers who fled the state to prevent quorum, impacting legislative processes. The arrests resulted in warrants that empowered law enforcement to detain lawmakers, though the warrants are symbolic due to jurisdiction limits. The absence of Democrats is being leveraged as a political tool to force their return and resume critical legislative work amid a session focused on redistricting, previously initiated due to significant Republican pressure. The situation reflects ongoing tensions associated with the state's political landscape and governance dynamics. (KRGV, August 4, 2025, https://www.krgv.com/news/texas-house-votes-to-track-down-arrest-democratic-lawmakers-who-left-the-state) | 0.17 | Civil arrest warrants were issued by the Texas House in early August 2025, but they were only enforceable within Texas; most absent Democrats remained out of state. Past precedent (e.g., 2021) suggests such warrants rarely result in actual detentions, and ongoing legal challenges further reduce the likelihood. While a brief detention upon return could count, the window before Sep 1, 2025 was short and incentives favored Democrats staying away. |

## A.6 CONTEXT SNAPSHOT GENERATION

To generate context snapshots, our system leverages a dual-strategy information retrieval pipeline.

The first method incorporates a search-augmented LLM, specifically `gpt-4o` and Grounding with Bing. We used the `gpt-4o-2024-08-06` snapshot. This setup performs live web searches and

Table 8: The prompt used for generating context snapshots with the search-integrated LLM, specifically `gpt-4o` and Grounding with Bing.

---

You are an expert superforecaster, familiar with the work of Philip Tetlock.

# Instructions
Given all you know, make the best possible prediction for whether each of these markets will resolve to Yes. Search the web for reliable and up-to-date information that can help forecast the outcomes of these markets. We expect you to answer in this format:

RESEARCH REPORT:
Write a *complete* record of the full search results (at least 5 paragraphs). Use plain text without markdown formatting.

---

Table 9: The prompt used for generating context snapshots with the custom RAG pipeline, specifically for search query construction.

---

The following are markets under the event titled "{event title}". The markets can resolve before the scheduled close date.

# Market 1
Title: {market title}
Subtitle: {market subtitle}
Possible Outcomes: Yes (0) or No (1)
Rules: {market primary rules}
Secondary rules: {market secondary rules}
Scheduled close date: {market expiration time}
(Note: The market may resolve before this date.)

# Market 2
...

# Instructions
What are 6 short search queries that would meaningfully improve the accuracy and confidence of a forecast regarding the market outcomes described above? Output exactly 6 queries, one query per line, without any other text or numbers. Each query should be less than 7 words.

---

synthesizes concise contextual summaries from up-to-date online content. The prompt used to guide the model in producing these snapshots is detailed in Table 8.

The second method employs our RAG pipeline. It begins with the use of `gpt-4o-mini`, which generates six context-specific search queries based on the prompt outlined in Table 9. These queries are then fed into the Dux Distributed Global Search (DDGS) library, which returns a curated list of relevant URLs along with their titles and brief descriptions.

Next, we scrape the content from each of the retrieved web pages. To distill this information, we again use `gpt-4o-mini`, this time prompting it (see Table 10) to produce concise summaries. The summarization prompt incorporates the page title, body, URL, and full scraped content. Each search query results in a single summary, yielding six summaries per event.

To maintain topical relevance, a filtering stage is applied to discard summaries that are off-topic or irrelevant to the original event. This step is handled by `gpt-5-mini`, guided by the prompt in Table 11. For `gpt-5-mini`, we used the `gpt-5-mini-2025-08-07` snapshot and for `gpt-4o-mini`, we used the `gpt-4o-mini-2024-07-18` snapshot.

## A.7 PROMPTS FOR THE GRANULARITY EXPERIMENT

In the prompt granularity experiment, we compare two levels of prompting: market-level prompting and event-level prompting. Each prompting strategy is evaluated both with and without the inclusion of context snapshots (i.e., research report excerpts). The market-level prompt with context snapshots is shown in Table 12. The version without context snapshots is identical, except that the research report content is omitted. Likewise, the event-level prompt with context snapshots is presented in Table 13, and the corresponding version without context snapshots simply excludes the research report excerpts. All prompts were used with both the `gpt-5-2025-08-07` snapshot and the `gpt-4o-2024-08-06` snapshot.

Table 10: The prompt used for generating context snapshots with the custom RAG pipeline, specifically for the summarization of the content of the relevant URLs.

```
The following are markets under the event titled "{event title}". The markets can resolve before the scheduled close date.

# Market 1
Title: {market title}
Subtitle: {market subtitle}
Possible Outcomes: Yes (0) or No (1)
Rules: {market primary rules}
Secondary rules: {market secondary rules}
Scheduled close date: {market expiration time}
(Note: The market may resolve before this date.)

# Market 2
...

# Article 1
Title: {article title}
Body: {article description}
Source URL: {article link}
Full Content: {article content}

# Article 2
...

# Instructions
Carefully read the articles provided above. Your task is to generate a multi-paragraph summary (one paragraph per article) that highlights factual insights
or relevant context related to the listed markets. Avoid subjective opinions or speculative statements. Use plain text without markdown syntax, headings, or
numbering. Do not add any additional text outside the summary.
Return blank for an article that does not contain relevant information. Not all of the articles are relevant to the markets above. Some are clearly unrelated to the
topic and should be excluded. Exclude only the articles that are clearly off-topic, entirely unrelated to the markets. If an article is at least broadly related or offers
potentially useful context, it should be considered relevant.
Important note: Include the date and source URL of the article at the end of each paragraph.
```

Table 11: The prompt used for the post-hoc filtering of the RAG context snapshots.

```
You are given a description of a prediction market, and 6 research reports generated to help predict the outcome of the market. However, not all of the reports are
relevant. Some are clearly unrelated to the topic and should be excluded. Your task is to identify only the reports that are clearly off-topic, those that are entirely
unrelated to the market. If a report is at least broadly related or offers potentially useful context, it should be considered relevant and not flagged. Carefully read
the market description and each report. Then, select the reports that are clearly irrelevant to the prediction task. The market and reports are:

The following are markets under the event titled "{event title}". The markets can resolve before the scheduled close date.

# Market 1
Title: {market title}
Subtitle: {market subtitle}
Possible Outcomes: Yes (0) or No (1)
Rules: {market primary rules}
Secondary rules: {market secondary rules}
Scheduled close date: {market expiration time}
(Note: The market may resolve before this date.)

# Market 2
...

# Research Report 1
{context snapshot 1}

# Research Report 2
...
```

## A.8 MIXED-EFFECTS REGRESSION RESULTS

We estimated a mixed-effects regression model to evaluate how model choice, prompting strategy, domain category, and the inclusion of context snapshots influence forecasting accuracy, measured by Brier score. Formally, the specification is given by

$$\text{Brier}_{ij} = \beta_0 + \beta_1 \text{Model}_i + \beta_2 \text{Strategy}_i + \beta_3 \text{Snapshot}_i + \beta_4 (\text{Model}_i \times \text{Strategy}_i) + \beta_5 \text{Domain}_i$$
$$+ \beta_6 (\text{Snapshot}_i \times \text{TimeToResolution}_i) + u_j + \epsilon_{ij}$$

where $\text{Brier}_{ij}$ denotes the Brier score for an observation $i$ on market $j$, $\text{Model}_i$ is a categorical variable with six levels (gpt-5, gpt-4o, claude-3.5-haiku, gemini-2.0-flash, llama-3.1-70B, and qwen-2.5-72B), where gpt-5 serves as the reference category. $\text{Strategy}_i$ is an indicator for market-level prompting (with event-level as the baseline), and $\text{Snapshot}_i$ indicates whether context snapshots were included or excluded (with inclusion as the baseline). $\text{Domain}_i$ is a categorical variable capturing the topic of the forecasting question (e.g., Sports, Pol-

Table 12: The market-level prompt used for the prompt granularity experiment.

```
# Market
Title: {market title}
Subtitle: {market subtitle}
Possible Outcomes: Yes (0) or No (1)
Rules: {market primary rules}
Secondary rules: {market secondary rules}
Scheduled close date: {market expiration time}
(Note: The market may resolve before this date.)

# Research Report 1
{context snapshot 1}

# Research Report 2
{context snapshot 2}

# Research Report 3
{context snapshot 3}

# Research Report 4
{context snapshot 4}

# Instructions
Given all you know, make the best possible prediction for whether this market will resolve to Yes. Format your prediction as a JSON object with the following
structure. There should be no text outside the object.
- "ticker": "KXWTAMATCH-25JUN30KALSTO" // market ticker copied exactly from the market metadata
- "reasoning": "A brief explanation of how you arrived at the prediction"
- "prediction": 0.00 // a probability between 0 and 1, inclusive.
```

Table 13: The event-level prompt used for the prompt granularity experiment.

```
The following are markets under the event titled "{event title}". The markets can resolve before the scheduled close date.

# Market 1
Title: {market title}
Subtitle: {market subtitle}
Possible Outcomes: Yes (0) or No (1)
Rules: {market primary rules}
Secondary rules: {market secondary rules}
Scheduled close date: {market expiration time}
(Note: The market may resolve before this date.)

# Market 2
...

# Research Report 1
{context snapshot 1}

# Research Report 2
{context snapshot 2}

# Research Report 3
{context snapshot 3}

# Research Report 4
{context snapshot 4}

# Instructions
Given all you know and the research reports above, make the best possible prediction for whether each of these markets will resolve to Yes. Format your
predictions as a JSON array of objects, where each object corresponds to a market. The length of your array must be {number of markets}. Include ALL
markets, even if you think they will resolve to No. There should be no text outside the array. Each object should have the following structure:
- "ticker": "KXWTAMATCH-25JUN30KALSTO" // market ticker copied exactly from the market metadata
- "reasoning": "A brief explanation of how you arrived at the prediction"
- "prediction": 0.00 // a probability between 0 and 1, inclusive.
```

itics, Finance), with Sports serving as the reference category. TimeToResolution$_i$ is a continuous variable defined only when snapshots are present; its effect is therefore modeled through the interaction Snapshot$_i$ × TimeToResolution$_i$. This variable represents the elapsed time between when the snapshot was taken and when the question ultimately resolved. The term $u_j \sim \mathcal{N}(0, \sigma_u^2)$ captures random intercepts at the market level to account for heterogeneity across markets, and $\epsilon_{ij} \sim \mathcal{N}(0, \sigma^2)$ represents the residual error.

The regression results show clear differences in forecasting accuracy across models, prompting strategies, and domains. Using gpt-5 as the reference category, all other models display significantly higher Brier scores, indicating worse accuracy on average. Prompting strategy exhibits no main effect for gpt-5, but significant interaction terms reveal that several models — specifi-

cally `gpt-4o`, `claude-3.5-haiku`, and `qwen-2.5-72B` — perform substantially worse under market-level prompting compared to event-level prompting. In contrast, `gemini-2.0-flash` and `llama-3.1-70B` show no reliable difference between the two prompting strategies.

Across all models, removing context snapshots leads to higher Brier scores, confirming that snapshot information improves predictive accuracy ($p < .001$). Domain effects reveal meaningful variation in task difficulty: Politics, Finance, and Climate & Weather questions are significantly harder to forecast than Sports, while Entertainment questions are easier. Finally, the interaction between snapshots and time-to-resolution is small and not statistically significant, indicating no reliable evidence that the time-to-resolution meaningfully affects forecasting accuracy.

| | Coefficient | Std. Error | $z$ | $p$ |
|---|---|---|---|---|
| Intercept | 0.184*** | 0.009 | 20.521 | $< .001$ |
| **Model (ref = GPT-5)** | | | | |
| GPT-4o | 0.016*** | 0.003 | 4.801 | $< .001$ |
| Claude 3.5 Haiku | 0.022*** | 0.003 | 6.586 | $< .001$ |
| Gemini 2.0 Flash | 0.015*** | 0.003 | 4.546 | $< .001$ |
| Llama-3.1-70B | 0.023*** | 0.003 | 6.787 | $< .001$ |
| Qwen-2.5-72B | 0.014*** | 0.003 | 4.082 | $< .001$ |
| **Strategy (ref = Event-level)** | | | | |
| Market-level | 0.003 | 0.003 | 0.757 | .449 |
| **Snapshots (ref = With snapshots)** | | | | |
| Without snapshots | 0.012*** | 0.003 | 4.893 | $< .001$ |
| **Domain (ref = Sports)** | | | | |
| Economics | 0.026 | 0.036 | 0.726 | .468 |
| Politics | 0.059** | 0.023 | 2.605 | .009 |
| Science & Technology | 0.052 | 0.045 | 1.164 | .245 |
| Entertainment | -0.043** | 0.014 | -3.005 | .003 |
| Finance | 0.170* | 0.078 | 2.196 | .028 |
| Climate & Weather | 0.110* | 0.049 | 2.239 | .025 |
| Health | 0.240 | 0.154 | 1.559 | .119 |
| **Model × Strategy** | | | | |
| GPT-4o × Market-level | 0.016** | 0.005 | 3.249 | .001 |
| Claude × Market-level | 0.014** | 0.005 | 3.001 | .003 |
| Gemini × Market-level | 0.005 | 0.005 | 0.947 | .344 |
| Llama × Market-level | 0.002 | 0.005 | 0.494 | .621 |
| Qwen × Market-level | 0.016*** | 0.005 | 3.420 | $< .001$ |
| **Time to Resolution** | | | | |
| With snapshots × TimeToResolution | -0.002 | 0.001 | -1.875 | .061 |
| Random intercept variance (Market Ticker) | 0.023 | 0.008 | | |

$^{***}p < 0.001;\ ^{**}p < 0.01;\ ^{*}p < 0.05$

Table 14: Mixed-effects regression of Brier score on model, prompting strategy, snapshots, domain category, and time-to-resolution, with random intercepts by market.

We conducted simple slopes analyses using estimated marginal means (EMMs) from the mixed-effects model. Pairwise contrasts compared event-level versus market-level prompting separately within each model, with Holm-adjusted $p$-values and asymptotic degrees of freedom.

As summarized in Table 15, results show clear model-dependent differences in sensitivity to prompt granularity. Event-level prompting significantly outperforms market-level prompting for `gpt-4o` (estimate $= -0.018$, $SE = 0.003$, $z = -5.35$, $p < .001$), `claude-3.5-haiku` (estimate $= -0.017$, $SE = 0.003$, $z = -5.00$, $p < .001$), and `qwen-2.5-72B` (estimate $= -0.019$, $SE = 0.003$, $z = -5.59$, $p < .001$). A smaller but still significant difference is observed for `gemini-2.0-flash` (estimate $= -0.007$, $SE = 0.003$, $z = -2.10$, $p = .036$). In contrast, the effects are nonsignificant for `gpt-5` (estimate $= -0.003$, $SE = 0.003$, $z = -0.76$, $p = .449$) and

`llama-3.1-70B` (estimate $= -0.005$, $SE = 0.003$, $z = -1.46$, $p = .146$), indicating that these models perform similarly under both prompting strategies.

Table 15: Simple slopes of prompting strategy within each model (event-level vs. market-level), based on estimated marginal means averaged over snapshot inclusion and domain.

| Model | Contrast | Estimate | SE | $z$ | $p$ |
|---|---|---|---|---|---|
| gpt-5 | Event − Market | −0.0026 | 0.0034 | −0.76 | .449 |
| gpt-4o | Event − Market | −0.0182 | 0.0034 | −5.35 | < .001*** |
| gemini-2.0-flash | Event − Market | −0.0071 | 0.0034 | −2.10 | .036* |
| claude-3.5-haiku | Event − Market | −0.0170 | 0.0034 | −5.00 | < .001*** |
| llama-3.1-70B | Event − Market | −0.0049 | 0.0034 | −1.46 | .146 |
| qwen-2.5-72B | Event − Market | −0.0190 | 0.0034 | −5.59 | < .001*** |

### A.9 Prompts for the Context Ensemble Effects

In the experiment investigating context ensemble effects, we compare two conditions: an Ensemble condition, which includes multiple context snapshots, and a No Ensemble condition, which includes a single snapshot. Both conditions utilize event-level prompting, and the prompt structure remains consistent with that shown in Table 13. The only difference lies in the number of research reports (i.e., context snapshots) included in the prompt—four in the Ensemble condition and one in the No Ensemble condition. All prompts were used with the `gpt-5-2025-08-07` snapshot.

### A.10 Sampling Procedure

From the 2,072 resolved questions with backtestable RAG snapshots collected as of September 22, 2025, we applied a series of filters to construct our experimental sample. These filters were chosen to ensure well-posed questions, stable metadata, and reliable resolution, all prerequisites for valid backtesting. The final dataset comprises 1,336 questions from 566 distinct events, with each question paired with four RAG snapshots.The filtering criteria were as follows:

- The question was published after the knowledge cutoff of `gpt-5` (September 30, 2024). This ensures that the model could not have directly learned the answer during training, thereby preventing temporal leakage and preserving the validity of the backtest.
- The event contains no more than six associated markets. This restriction prevents extremely large events (e.g., those with 50+ markets) from disproportionately influencing the analysis.
- Market prices at the simulated prediction time (i.e., the snapshot generation time) are available. Price availability is necessary for computing forecasting accuracy and all downstream evaluation metrics.
- The question has a definitive resolution outcome ("yes" or "no"). Binary outcomes are required to evaluate probabilistic predictions in a consistent manner.
- The question did not resolve before the simulated prediction time, even if the official market close date was later. (See the next section for details on these exclusions.)
- At least four RAG snapshots are available for the question at the simulated prediction time. A minimum of four RAG snapshots is required to allow the ensembling setup.

Applying the same filtering procedure to the Bing snapshots produced a comparable experimental sample. From 1,435 resolved questions with backtestable Bing snapshots, 779 questions remained after filtering, spanning 340 unique events.

### A.11 Eliminating Edge Cases of Temporal Contamination

To ensure that LLMs are genuinely forecasting future events rather than recalling known outcomes, we introduced an additional filtering step. Specifically, we excluded any prediction markets where the official close time occurred after the actual resolution of the underlying event.

For example, consider a market about the outcome of the Cincinnati vs. Philadelphia MLS soccer match, with the rule: "If Philadelphia wins the Cincinnati vs. Philadelphia professional MLS soccer game originally scheduled for August 30, 2025, after 90 minutes plus stoppage time (excluding extra time or penalties), then the market resolves to Yes." Although the event resolves at the end of regular time on August 30, the market's official close time is listed as August 31, 2025, at 01:41:33 AM. This means that a simulated prediction made on August 30—intended to be prior to the event—could actually occur after the game's outcome is already known, but before the market officially closes.

To eliminate such edge cases, we processed all candidate markets using `gpt-5-nano`, filtering out any where the official close time trailed the real-world event resolution. Specifically, we used the `gpt-5-nano-2025-08-07` snapshot. This ensured that our final experimental dataset was free from temporal contamination and consisted only of markets where true forecasting was required.

### A.12 Validation of Information Source and Relevance

To validate the information sources for the context snapshots, we verified every URL used in both pipelines. For the Bing snapshots, we checked all URLs returned by Grounding with Bing during

snapshot generation. For the RAG snapshots, we validated all URLs retrieved through the DDGS library that contributed to the final snapshots. Every URL was processed through the Google Safe Browsing API to identify any indicators of non-credible or unsafe websites. No URLs were flagged, meaning all sources used to generate our context snapshots passed the safety screening.

In terms of relevance, the RAG pipeline includes a filtering and ranking step that scores HTML content using cosine similarity, ensuring that only relevant text is summarized into RAG snapshots. For the Bing snapshots, we rely on Grounding with Bing and the GPT-4o model to generate content that is already optimized for relevance. As an additional verification step, we computed the cosine similarity between the embeddings of each context snapshot and the associated market metadata. We used the all-mpnet-base-v2 model from sentence-transformers library as the embedding model. The market metadata included the event title (the question being asked), market title, market subtitle, primary rules, and secondary rules. All snapshots achieved a similarity score of 0.3 or higher. These relevance scores are included in the released dataset under the "relevance" field.

