# OpenReview forum: "Forecasting with LLMs: A Dataset for Rapid Backtesting Without Temporal Contamination"
_ICLR.cc/2026/Conference — Submitted to ICLR 2026_

### Official Review · Reviewer_PYP1 · 2025-10-30

**Soundness:** 2
**Presentation:** 1
**Contribution:** 3
**Rating:** 4
**Confidence:** 3

**Summary:**

The paper introduces a replayable, leakage-resistant LLM forecasting benchmark that pairs real yes/no prediction-market questions with frozen, time-stamped web summaries captured before resolution. It shows snapshots (and small ensembles) improve Brier accuracy; event-level prompting helps GPT-4o but not GPT-5; markets remain a strong baseline.

**Strengths:**

* Freezing dated web evidence to replay forecasts later is a simple idea that cleanly separates model reasoning from training-data freshness and search, yielding an immediately useful benchmark across domains.
The setup is careful and rigorous: clear timestamps, transparent snapshot pipelines, explicit prompts, Brier scoring, and a strong market-price baseline; results are sensible (snapshots help, small ensemble gains).
* The benchmark upon release will have practical benefit to the community.

**Weaknesses:**

* The paper doesn’t isolate “reasoning” from plain prediction: we never see whether deliberate modes (e.g., CoT, Thinking models) actually convert the same snapshots into better forecasts than a straightforward probability output.
* There’s no apples-to-apples comparison with tool-using systems. The benchmark forbids browsing, but many state-of-the-art forecasters rely on tools/search; we don’t see the same model with tools vs the same model restricted to frozen notes.
* The task space is binary only but many real forecasts are multi-way or continuous, so external validity is limited.
* Some experimental flaw: 1) event-level prompts may help simply because they’re longer. The study doesn’t control for token budget, so framing improvements are confounded with “more text.” 2) Frozen notes can contain quoted odds or implied probabilities. Models might piggyback on crowd signals rather than reason from evidence, and there’s no redaction/ablation to rule this out.
* Comparability will drift over time: to remain forward-only, newer models require a newer evaluation window (answer might be contained in training data), which forces re-running baselines on that window and prevents a single, consistent, time-stable leaderboard.
* The deeper analysis on the benchmark evaluations is lacking: 1) baselines are narrow that results are centered on a small set of closed models, with no strong open-model or classical forecasting baselines to gauge generality. 2) domain and time-horizon analysis is shallow that results aren’t systematically broken down by question type, difficulty, or days-to-resolution, so we can’t see where the method really helps or fails.
* Text in figures are very small (espeically figure 3), and zooming in makes it blurry.

**Questions:**

* The code link is provided but inaccessible.
* Can you share 2–3 real benchmark examples (like A.3) paired with actual model outputs?
* Why not just limit Google Search to a cutoff date instead of using frozen snapshots?

---

> ### Author Response · Authors · 2025-11-28
>
> **W1. Isolation of reasoning capability**
>
> We agree that isolating reasoning from straightforward prediction is an important research direction. Our goal in this paper, however, is to show that the dataset enables such investigations rather than to exhaustively explore all possible strategies. There is a broad space of prompting methods, deliberate-reasoning modes, ensembling techniques, and more, and our framework is designed precisely to make these comparisons feasible. There are countless strategies and countless subsets of the dataset that could be interesting to evaluate. In this work, we focused on prompt granularity and context-ensembling experiments to illustrate the dataset’s utility, but the same pipeline can support future studies on reasoning-specific effects. Our benchmark dataset makes it possible to hold every factor constant and conduct an instant backtest of reasoning gains.
>
> Please also see our general response point P6.
>
> **W2. Comparison with tool-using systems**
>
> We appreciate the reviewer’s point that our benchmark does not capture the comparison with tool-using systems. Our goal, however, is not to evaluate tool-use per se but to enable methodologically clean, apples-to-apples comparisons of forecasting strategies under identical information conditions. By freezing contemporaneous context, we isolate the contribution of the forecasting method rather than the quality or recency of external search. This design choice intentionally excludes live browsing so that strategies (e.g., base-rate, reference-class, or model-based) can be compared without temporal contamination or variability in tool performance. We have updated Section 3.2 to clarify this distinction and to explain why tool-use comparisons, though important, fall outside the intended scope of our framework.
>
> Please also see our general response point P6.
>
> **W3. Using binary questions**
>
> We focused on binary questions because our primary goal was to enable clean, cross-domain comparisons. Mixed outcome types introduce confounds: for example, multi-way questions differ in their base rates and number of options, making it difficult to make apples-to-apples comparisons. Binary questions avoid these issues and provide a consistent evaluation substrate.
>
> Moreover, if a key use case of the benchmark is comparing forecasting strategies to human performance on the same questions, binary formats are convenient. Prediction markets fundamentally operate on binary contracts: even continuous quantities (e.g., interest rates) are decomposed into multiple binary markets. Thus, the most reliable and widely available source of human “wisdom of the crowd” exists in binary form.
>
> That said, extending the framework to multi-way or continuous outcomes is entirely feasible; we simply prioritized comparability for this initial release.
>
> **W4. Experimental flaws**
>
> *On the “longer prompts” alternative explanation.*
>
> Our results suggest that the performance gap is not driven merely by prompt length. Prior work has shown that longer prompts do not reliably improve accuracy, and in fact can degrade performance when additional text diffuses salient information (Du et al., 2025). Our findings are consistent with this literature. If the advantage were primarily a function of token count, we would expect the effect to generalize across models. Instead, we observe a strong benefit for event-level prompting in gpt-4o, but the effect vanishes entirely for gpt-5, even though both models receive similarly sized prompts.
>
> Du, Yufeng, et al. "Context length alone hurts LLM performance despite perfect retrieval." Proceedings of the 2025 Conference on Empirical Methods in Natural Language Processing. 2025.
>
> *On snapshots containing quoted odds or implied probabilities.*
>
> We appreciate the concern that snapshots may contain quoted odds or implied probabilities that models could exploit. Our aim with this dataset, however, is not to optimize absolute forecasting performance but to provide a controlled, historically frozen environment for comparing strategies and models under identical information sets. If a model uses market-implied probabilities, that is itself a meaningful behavioral difference, and our framework enables researchers to study such effects systematically (e.g., by comparing performance against a redacted set of context snapshots). Importantly, the relative comparisons we report remain valid regardless of whether quoted odds appear in some snapshots, because every evaluated strategy operates over the same frozen context.
>
> **W5. Comparability will drift over time**
>
> Please see our general response point P3.

---

> ### Author Response · Authors · 2025-11-28
>
> **W6. Narrow baselines and Additional Analyses**
>
> Regarding the narrow baselines, please see our general response point P2.
> Regarding the additional analyses, please see our general response point P8.
>
> **W7. Figure text and resolution**
>
> We have addressed this issue by regenerating all figures with larger, higher-resolution text to ensure clarity without blurriness when viewed or zoomed.
>
> **Q1. Anonymous link**
>
> Please see our general response point P7.
>
> **Q2. Benchmark examples paired with actual model outputs**
>
> We added Table 7 in Appendix A.5, which shows examples of RAG snapshots from our dataset, paired with the corresponding GPT-5 predictions and reasoning, each produced using only a single snapshot as input.
>
> **Q3. Google search with date limit**
>
> Please see our general response point P5.

---

### Official Review · Reviewer_Tn4L · 2025-10-31

**Soundness:** 3
**Presentation:** 3
**Contribution:** 2
**Rating:** 4
**Confidence:** 4

**Summary:**

The paper proposes a backtestable forecasting strategy using frozen context snapshots. These are timed summaries of web search results paired with prediction market questions. Their goal is to avoid any temporal contamination and reduce any stale confounding factors. And the pipeline itself is a two-stage pipeline. The first step being the search integrated LLM and then there is a custom RAG system. The experiments show that the snapshots improve prior scores, event level prompting beats market level prompting for GPT-4o but not for GPT-5, and that the ensemble snapshots yield small domain-dependent gains.

**Strengths:**

1. The forward-only snapshots directly tackle leakage and reduce staleness, which is a quite clear evaluation target. The dataset contains over 9000 questions with over 3000 results. There is sufficient context captured after model training cutoffs which enables back testing without contamination.

2. There are two complementary retrieval pipelines with daily timestamped runs.

3. The Breyer metric and the mixed effect regression and the event versus market prompting and ensemble of applications are quite concrete and interpretable.

4. There is a practical release plan with legal aware summarization instead of raw content as well.

**Weaknesses:**

1. The core idea of the research and freezing context is quite straightforward. It's more of a benchmark paper, With the retrieval methods as quite standard, which is Bing search and basic RAG.

2. The context ensembling methods show minimal gains and the prompt granularity findings don't really generalize and there is no compelling evidence that the approach is significantly improving forecasting.

3. There is a missing analysis on which types of questions which benefit from context. Summarization may lose critical details, especially if done using LLMs.

4. The post hop filtering of unrelated summaries using GPT-5 mini seems a bit circular. There is no validation of information source, quality or relevance.

5. All the questions being from Kalshi limit the source to a single source which limits generalizability. The filtering criteria in the appendix seems arbitrary and lacks justification. And the temporal gap between the 2025 July snapshot and the 2024 September model cutoff still leaves some room for contamination through training on related historical patterns.

**Questions:**

1. How do you validate that the snapshots contain decision-relevant information?

2. What is the maintenance plan as the data set ages, if any at all.

3. How does the performance vary with time to resolution and can you please provide evidence that summarization will preserve critical information and what is your plan for that?

---

> ### Author Response · Authors · 2025-11-28
>
> **Summary**
>
> We would like to clarify a misunderstanding in the reviewer’s summary. Our system is not a two-stage pipeline; rather, it consists of two independent pipelines. One uses a search-integrated LLM (Grounding with Bing), and the other uses a custom RAG approach. The two types of context snapshots are therefore generated independently, not sequentially. We have revised the paper to improve readability and to make this distinction clearer throughout.
>
> **W1. Contribution of the Benchmark**
>
> We appreciate the reviewer’s comments and the opportunity to clarify our contributions. Temporal contamination and staleness confounds remain as persistent issues in the existing literature. While the high-level idea of freezing context may seem straightforward, our work provides the first fully implemented, forward-only, backtestable evaluation framework designed to eliminate these two issues. The contribution is not only conceptual but also practical: we develop an end-to-end pipeline that continuously collects unresolved prediction market questions, captures contemporaneous structured context, and enables rapid, reproducible backtesting once outcomes resolve. To our knowledge, no existing benchmark offers this combination of properties. Regarding retrieval, we agree that using Grounding with Bing and our custom RAG pipeline are standard. The novelty lies not in the individual retrieval primitives but in how we operationalize them for forecasting evaluation. We hope this clarifies the significance and distinctiveness of our work.
>
> **W2. No Evidence that the Approach Significantly Improves Forecasting Accuracy**
>
> Our main argument is not that any single strategy, such as context ensembling or prompt granularity, significantly improves the forecasting accuracy. Rather, these experiments serve as demonstrations of how our forward-only, backtestable framework enables rapid iteration and test of diverse forecasting strategies, which we argue is important for discovering what has the potential to improve performance. That is, the contribution of the paper is the methodology and dataset that make such rapid, leakage-free backtesting possible, not the magnitude of gains from the specific strategies we happened to test. We have revised the wording accordingly to make this clearer and have toned down claims regarding the observed improvements. Thank you for pointing this out.
>
> Please also see our general response point P6.
>
> **W3. Missing analysis & loss of information due to summarization**
>
> Regarding the missing analysis on question types, please see our general response point P8.
>
> Regarding the loss of information due to summarization, please see our general response point P1.
>
> **W4. Validation of information source, quality, and relevance**
>
> Thank you for the reviewer’s thoughtful comment. We agree that source quality is essential for ensuring the credibility of context snapshots. Our framework already incorporates two safeguards that mitigate low-quality or irrelevant inputs: (1) the use of search-integrated LLM (Grounding with Bing) and DDGS (Dux Distributed Global Search), both of which prioritize high-authority pages in their ranking algorithms, and (2) a post-hoc filtering stage that automatically removes snapshots that are clearly off-topic or inconsistent with the focal event using LLM-as-a-judge. These steps ensure that only the contextually relevant snapshots are retained. In addition, RAG snapshots undergo a semantic-similarity check in which each web page retrieved from the DDGS search library is compared to the market metadata (e.g., title, rules). We use this similarity score to rank and filter pages before summarization, further strengthening the quality and relevance of the resulting context.
>
> To further address the reviewer’s concern, we added a post-hoc source credibility analysis to our paper. For each URL used in a context snapshot, we compared its domain against the Google Safe Browsing API to assess its credibility. Additional details are provided in Appendix A.11.

---

> ### Author Response · Authors · 2025-11-28
>
> **W5. Justification of Kalshi as the source and filtering criteria**
>
> As mentioned in Section 6.1, we acknowledge that our current dataset is sourced exclusively from Kalshi. Kalshi is one of the largest real-money forecasting platforms, characterized by high question volume and broad topical coverage, making it a strong initial testbed. Specifically, Kalshi reports more than $50 billion traded annually and has recently surpassed Polymarket in total trading value (Yahoo Finance, 2025; Kalshi Data, 2025). Moreover, most prediction platforms share a similar trading structure to Kalshi, and Kalshi spans a wide range of domains, including sports, politics, economics, and global events, which substantially overlap with other leading prediction markets. In addition, we may expand it in future work to include additional platforms to improve generalizability.
>
> We have clarified in Appendix A.10 that the filtering criteria are not ad hoc; they are chosen to ensure well-posed questions, stable metadata, and reliable resolution, all prerequisites for valid backtesting.
>
> Finally, the temporal gap between model cutoff and snapshot does not introduce leakage: the model’s accessible information is limited to its pre-cutoff training data and the timestamped context snapshot, both of which precede resolution. We generate the context snapshot for active events and then wait for those events to resolve. Thus, at the time the snapshot is produced, no temporal leakage can occur. Please let us know if we have misunderstood the reviewer’s concern.
>
> Yahoo Finance. (2025). Prediction Market Kalshi’s Valuation Jumps to \$11B After Reported \\$1B Raise. https://finance.yahoo.com/news/prediction-market-kalshi-valuation-jumps-130434026.html
>
> Kalshi Data. (2025). Kalshi trading statistics. https://www.kalshidata.com
>
> **Q1. Validating that snapshots contain decision-relevant information**
>
> As noted in our response to W4, our framework already includes safeguards that ensure snapshots contain decision-relevant information. First, search-integrated LLM and DDGS prioritize high-quality, authoritative sources. Second, a post-hoc LLM-as-a-judge filter removes off-topic or inconsistent snapshots. For RAG snapshots, each retrieved page undergoes a semantic-similarity check against the market metadata, ensuring that only contextually relevant content is summarized. Together, these steps bring the snapshots closer to reflecting the information a forecaster would reasonably consult at the time.
>
> **Q2. Maintenance plan**
>
> Please see our general response point P3.
>
> **Q3. Additional Analyses**
>
> Please see our general response point P8.

---

### Official Review · Reviewer_Zrzm · 2025-11-01

**Soundness:** 2
**Presentation:** 3
**Contribution:** 3
**Rating:** 6
**Confidence:** 4

**Summary:**

This paper presents a pipeline for building LLM forecast benchmarks. It continuously scrapes unresolved questions from the prediction market Kalshi. The framework prevents temporal contamination by capturing supporting context at the time of question scraping using two methods.

**Strengths:**

- The proposed framework effectively addresses temporal contamination and evaluation-delay issues.
- The benchmark can be valuable to the community if it is kept up to date.

**Weaknesses:**

- This is primarily a benchmark paper. That being said, it lacks substantial methodological or theoretical novelty. It is fine as a benchmark contribution, but I think it fits better at benchmark-focused venues, such as the NeurIPS DB track.
- It would be better to evaluate more models and present their results on the proposed benchmark to provide more insights. Currently, the paper only covers GPT-4o and GPT-5, so it’s unclear how well the approach generalises to other proprietary and open-sourced models.
- The link to the dataset in the abstract doesn't work on my side.

**Questions:**

See those in the Weaknesses part.

**Details Of Ethics Concerns:**

I'm not sure whether scraping questions from Kalshi and related snapshots on the Internet would violate copyright. The authors provide summaries of the snapshots, but I'm not completely sure this eliminates all copyright concerns.

---

> ### Author Response · Authors · 2025-11-27
>
> **W1. Benchmark contribution**
>
> We thank the reviewer for their feedback. Our submission is a benchmark-focused contribution, targeted at the ICLR “datasets and benchmarks” core subject area (https://iclr.cc/Conferences/2026/CallForPapers). Recent ICLR editions have featured multiple benchmark papers that advanced evaluation standards without introducing new algorithms. Our work follows this tradition: it aims to contribute a dataset that fills a gap that is impeding progress in the rapidly growing area of LLM forecasting. We believe our contribution provides substantial value to ICLR researchers, and we hope the reviewer will reconsider the suitability of the venue in light of ICLR’s stated scope.
>
> **W2. Generalization to other proprietary and open-sourced models**
>
> Please see our general response point P2.
>
> **Ethics Review**
>
> Please see our general response point P4.

---

### Official Review · Reviewer_7ykK · 2025-11-01

**Soundness:** 2
**Presentation:** 2
**Contribution:** 3
**Rating:** 4
**Confidence:** 5

**Summary:**

The paper proposes alleviating concerns of leakage in forecasting backtests from context retrieval by releasing a fixed dataset of relevant snapshots to Kalshi forecasting questions. It collects these snapshots in two ways: 1) gpt-4o with bing search, 2) gpt-4o-mini generated summaries of DDGS search results to gpt-4o-mini generated queries. The retrieved articles lead to an improvement in brier score compared to no retrieval.

**Strengths:**

1. The idea of using fixed snapshots for retrieving information for forecasting backtests is important and useful.

2. Providing a dataset with past retrieved articles makes it one step less complex to do research on for eg model training for forecasting.

3. The questions are scraped from Kalshi, and their relevant context is retrieved automatically. Since the whole pipeline is automated, it is easy to update the data.

4. The experiment on context ensembling was interesting, though it is not fleshed out.

5. The appendix includes interesting details and examples.

**Weaknesses:**

1. The main weakness of the proposed dataset is that it loses information. Specifically, only a small, highly processed summary of the "snapshot" is provided as context. The summary provided is created by definition by a weaker, old model, in this case gpt 4o / mini. No clear quality comparisons are provided with alternatives to show how good the retrieved articles are. Due to this limitation, I would personally not use this dataset for forecasting research.

2. The methodology (providing a fixed set of past retrieved articles) is only helpful when the prediction is treated in isolation assuming a fixed input. However, seeking relevant information is an important step for forecasting, where more intelligence and reasoning help. This is also inherently a model capability, which one might train for during forecasting training. I would significantly rewrite 3.2 in light of this.

3. Directly providing access to source articles as done by commoncrawl or wayback, is far more flexible. I find the limitations of these alternatives mentioned in Appendix A.2 unconvincing. While commoncrawl loses some information as websites like CNN do not license scraping, the proposed method is likely to lose even more information. In fact, it is unclear how this method can provide such news articles legally. Similarly, while wayback is slow, I still believe a lot more context can be gathered from wayback than provided by this fixed pipeline.

**Questions:**

The anonymous url link is broken. Bringing it to your notice if this was unintentional :)

The critique of Wilderman et al (2025) on L163 is unclear to me. Why is it noisy or undesirable to use archives of google search?

In L173 the claim is made that context snapshots mitigate the issue of differing training cutoffs. While I generally agree with the claim, it would be good to see more evidence on how much context can overcome models with older cutoff dates.

I found the terms "event-level" and "market-level" a bit unintuitive to distinguish. I get the implied intent after reading 5.2.2, but this comes too late. Perhaps consider more intuitive terminology.

**Details Of Ethics Concerns:**

The paper claims that if web content is prohibited for redistribution, using the gpt generated summaries released in their dataset mitigates this issue. I am skeptical of this claim, and believe the proposal could still be illegal or unethical. I am not a legal expert, so it would be good to get this checked.

---

> ### Author Response · Authors · 2025-11-27
>
> **Summary**
>
> We appreciate the reviewer’s time and thoughtful evaluation of our paper. We would like to clarify one point regarding the summary that our work involves “releasing a fixed dataset of relevant snapshots to Kalshi forecasting questions.” In fact, the dataset is not fixed but dynamic. Our pipeline continuously collects active questions from the prediction market and saves the corresponding context snapshots. As questions resolve, we plan to release updated versions of the dataset on a regular basis at least through the end of 2026.
>
> **W1. Loss of information due to summarization**
>
> Please see our general response point P1.
>
> **W2. Fixed inputs not capturing retrieval ability**
>
> Please see our general response point P6.
>
> **W3. Comparison to CommonCrawl and Wayback**
>
> We agree that large-scale archives such as CommonCrawl and the Internet Archive’s Wayback Machine are invaluable resources, but we believe they are not practical substitutes for the purpose of forecasting evaluation, for several reasons.
>
> First, neither CommonCrawl nor Wayback supports context retrieval via keyword-based search, making it impossible to query efficiently. Because these archives require URL-based access, it is difficult to use them to construct a reproducible pipeline that retrieves contemporaneous, thematically relevant material for each forecasting question.
>
> Second, these archival resources are also computationally impractical for scaled backtesting. Both require time-intensive and brittle scraping pipelines to retrieve and parse raw HTML. This makes them unsuitable for repeated backtesting runs over thousands of forecasting questions, as is required to compare diverse forecasting strategies or models. Our frozen snapshots are lightweight and immediately usable, enabling faster research cycles.
>
> Finally, we view our framework as complementary, not competitive, with large archives. If future researchers wish to augment our snapshots with richer archival context, they can do so easily because our dataset already provides the question, query, and timestamp needed to anchor such searches. Our contribution is not intended to replace these archives but to offer a clean, leakage-free, forecasting-specific layer on top of the underlying information ecosystem.
>
> In short, while CommonCrawl and Wayback are excellent general-purpose archives, they do not solve the practical reproducibility required for large-scale, forward-only forecasting evaluation. Our framework fills that gap. We have accordingly updated Appendix A.1.3.

---

> ### Author Response · Authors · 2025-11-27
>
> **Q1. Anonymous Link**
>
> Please see our general response point P7.
>
> **Q2. Critique of Wilderman et al (2025)**
>
> Please see our general response point P5.
>
> **Ethics Review**
>
> Please see our general response point P4.

---

### Author Response · Authors · 2025-12-01
**General Response**

We have responded to each reviewer individually and uploaded the revised version of the manuscript. Additionally, we note that a few concerns were shared by more than one reviewer, and address them below:

**P1. Loss of information due to summarization**

We appreciate the reviewers’ concern regarding potential information loss due to the use of contemporaneous summaries rather than full articles. However, we believe that in practice this limitation is mitigated by the structure of our dataset and the objectives of our evaluation framework.

While we acknowledge that the limitations highlighted by the reviewer may affect the **average, absolute** level of performance achieved by any given approach in predicting the outcomes included in the dataset, we note that the value of the benchmark dataset is in allowing future researchers to assess the **relative** performance of different forecasting strategies. In other words, conditional on having access to the same context snapshots, which approaches best utilize the context snapshots (despite any limitations they may have) to make forecasts about unseen events?

That said, it is desirable to have the context snapshots be reasonable summaries of the web content that a search-enabled LLM or web search would retrieve for the event and date in question. For this reason, we create multiple contemporaneous context snapshots on a daily basis and store them for testing forecasting strategies later. Specifically, for each event we create and store an average of 4.08 “Grounding with Bing” snapshots and 5.11 RAG snapshots (Table 4 in the Appendix).

**P2.  Generalization to other proprietary and open-sourced models**

We appreciate the reviewer’s suggestion and have expanded our evaluation accordingly. Specifically, we extended the Prompt Granularity Experiment (Section 5.2) and incorporated updated results. The revised experiment now includes six models rather than the original two: GPT-4o, GPT-5, Claude 3.5 Haiku, Gemini 2.0 Flash, Llama-3.1-70B, and Qwen-2.5-72B. This broader set spans both proprietary and open-source model families, offering a more comprehensive view of how our framework generalizes across model classes.

We intentionally selected models whose training cutoffs fall within a comparable window to avoid temporal leakage. Using the newest releases, such as Claude 4.5 or Gemini 3.0, would introduce additional training data that extend closer to the snapshot dates. Therefore, we chose Claude Haiku 3.5 (training cutoff: July 2024), Gemini 2.0 Flash (training cutoff: Aug 2024), GPT-5 (training cutoff: Sep 2024), and GPT-4o (training cutoff: Oct 2023). For open-source baselines, we selected Llama-3.1-70B (training cutoff: Dec 2023) and Qwen-2.5-72B (training cutoff: end of 2023).

**P3. Maintenance plan**

Our dataset is continuously updated with newly scraped questions, ensuring it remains current. We guarantee active maintenance through at least the end of 2026, and will reflect this commitment in the acknowledgements upon publication.

**P4. Legal and ethical compliance**

We appreciate the reviewers’ attention to legal and ethical compliance. We note the following. We propose to distribute abstractive summaries (context snapshots), not copyrighted source materials. The OpenAI terms of use (https://openai.com/policies/terms-of-use/) state “Ownership of content. As between you and OpenAI, and to the extent permitted by applicable law, you (a) retain your ownership rights in Input and (b) own the Output. We hereby assign to you all our right, title, and interest, if any, in and to Output.” We will excise from the output any precise statements recommended to excise by our institutional review body. Review will be finalized immediately before publication because, as an ongoing benchmark, collection of context snapshots will continue until publication and yet uncollected snapshots cannot be reviewed. For example, OpenAI terms could change in the future and we would need to react accordingly.

---

> ### Author Response · Authors · 2025-12-01
>
> **P5. Google search with date limit**
>
> We have clarified why using Google Search can lead to temporal leakage in Appendix A.1.4 as follows.
>
> > Limiting Google search to a cutoff date does not fully address temporal contamination because search engines continually re-rank and retroactively update pages. Even if results are filtered by date, snippets, metadata, and page contents frequently reflect information added after the nominal cutoff, and many pages are undated or incorrectly dated. As a result, date-filtered live search is still exposed to leakage, making the evaluation neither reproducible nor temporally secure.
> >
> > Bench to the Future (Wildman et al., 2025) acknowledges that using live Google searches introduces the potential for information leakage based on the fact that the order of certain search results likely changed over time.
> >
> > Frozen snapshots solve these problems by capturing information as it appears at the time of query (which occurs before event resolution), independent of later edits, page removals, or algorithmic changes (which may occur after an event is resolved). This creates a stable and contemporaneous record that supports controlled backtesting.
>
> **P6. Clarification on the scope**
>
> Our point is to provide researchers with a rigorous way to evaluate different LLM-based forecasting techniques (e.g., base-rate methods, reference-class approaches, ensemble approaches) using a retrospective archive that is explicitly free of temporal contamination. The reviewer is correct that information seeking is an important component of real-world forecasting, and that models could be trained to improve this capability. However, we feel that question lies outside the scope of our work. Our framework is designed to hold the available information fixed so that different forecasting strategies can be compared relative to one another under identical conditions. We therefore do not focus on absolute forecasting performance but rather on enabling clean, reproducible **relative** comparisons across methods and models, without the confounds of temporal contamination or staleness. The static nature of the context snapshots is what allows us to address these confounds. As the reviewer suggested, we have revised Section 3.2 accordingly to clarify this distinction and to better emphasize the intended scope of our framework.
>
> > It is important to clarify the context our benchmark was designed to address. With regards to the model used for forecasting, this benchmark is designed to evaluate systems based on LLMs that (a) have training data cutoffs that precede the events used for benchmarking; and (b) do not access the live web during inference, which would introduce temporal contamination and invalidate the key advantage of this dataset. While models that are specifically trained to search for forecasting-relevant information may possibly improve forecasting performance, they lie outside the intended scope of systems to be evaluated by this benchmark.
>
> **P7. Anonymous link**
>
> We have fixed the anonymous link. It now contains the descriptive statistics of the context snapshots and examples of the snapshots for review purposes. As the set of context snapshots will grow until the publication date and internal review board approval cannot be finalized until the set is complete, we will publicly release the full dataset, minus any precise phrases we are advised to exercise, in the month prior to publication.
>
> **P8. Additional analyses**
>
> We have expanded our analyses to incorporate both the domain of each forecasting question and the time to resolution (i.e., time between the snapshot and the market’s resolution.) All results reported in Section 5 and Appendix A.8 now reflect this updated mixed-effects regression framework, and we have included the revised manuscript with these changes incorporated.

---

### Meta-Review · Area_Chair_oNZL · 2026-01-06

**Summary:**

The submission "Forecasting with LLMs: A Dataset for Rapid Backtesting Without Temporal Contamination" proposes a benchmark for forecasting algorithms constructed by continuous scraping of unresolved kalshi questions, which are stored together with summaries of bing search results from the same time period. Later, forecasting algorithms can then be backtested without contamination by evaluating the provided questions with the given summarized content.

**Reviewer Concerns:**

The main concern raised by the reviewers is that it is unclear whether the design of this benchmark is too constraining for future forecasting systems. The design removes the ability for models to use research on their own terms, and to synthesize information directly from source documents, as only summaries of web search results mediated by (by the time of benchmarking older) models, are provided. Reviewers argue that "Due to this limitation, I would personally not use this dataset for forecasting research". The authors respond in the rebuttal to argue that this benchmark would still allow for a relative comparison of forecasting strategies. However, this does not address that the space of potential forecasting strategies is still limited by the design of the benchmark, and only different strategies of understanding the predetermined context can be compared. Tool-using systems in general, for example, are ruled out, even though reviewers highlights that  "The benchmark forbids browsing, but many state-of-the-art forecasters rely on tools/search" and that "seeking relevant information is an important step for forecasting, where more intelligence and reasoning help. This is also inherently a model capability, which one might train for during forecasting training".

From a broader point of view, I personally also do not fully see why offline search/retrieval over already dumped internet corpora like CommonCrawl is not a more straightforward solution to the backtesting problem in forecasting. Commoncrawl dumps are guaranteed to be uncontaminated, are dumped automatically without the need for a new system that requires future maintanance and can already be used to backtest decades into the past, if desired. Further, documents are available directly in their source html-form and common internet parsing libraries can still be used to simplify the documents if desired (the rebuttal notes this as a concern with common crawl). The authors argue that offline search over CommonCrawl is expensive, but compared to compute and storage for the models employed in this work, search is computationally very efficient.





There were further a few minor concerns regarding the baseline methods tested in the submission, post-hoc filtering strategy and source diversity, which were partially addressed during rebuttal.

Reviewers also raised a flag for ethics review due to concerns about data copyright, but given the research angle of this benchmark and the fact that only summarized documents are provided, I do not see this as a concern for ICLR. Although the authors may want to check the Kalshi licensing details.


In summary, reviewers have convinced me that the current construction of this benchmark seems to be too constraining to really develop novel forecasting systems in the future and I am not recommending acceptance of this submission.

**Reviewer Scores:**

7ykK,Zrzm,Tn4L no change in score due to sustained conceptual concerns. PYP1 could increase to 6 based on rebuttal comments regarding baseline experiments.

---

### Decision · Program_Chairs · 2026-01-26

Reject